# EFFICIENT DATA SUBSET SELECTION TO GENERALIZE TRAINING ACROSS MODELS: TRANSDUCTIVE AND INDUCTIVE NETWORKS

## ABSTRACT

Subset selection, in recent times, has emerged as a successful approach toward efficient training of models by significantly reducing the amount of data and computational resources required. However, existing methods employ discrete combinatorial and model-specific approaches which lack generalizability— for each new model, the algorithm has to be executed from the beginning. Therefore, for data subset selection for an unseen architecture, one cannot use the subset chosen for a different model. In this work, we propose SUBSELNET, a non-adaptive subset selection framework, which tackles these problems with two main components. First, we introduce an attention-based neural gadget that leverages the graph structure of architectures and acts as a surrogate to trained deep neural networks for quick model prediction. Then, we use these predictions to build subset samplers. This leads us to develop two variants of SUBSELNET. The first variant is transductive (called as Transductive-SUBSELNET) which computes the subset separately for each model by solving a small optimization problem. Such an optimization is still super fast, thanks to the replacement of explicit model training by the model approximator. The second variant is inductive (called as Inductive-SUBSELNET) which computes the subset using a trained subset selector, without any optimization. Most state-of-the-art data subset selection approaches are adaptive, in that the subset selection adapts as the training progresses, and as a result, they require access to the entire data at training time. Our approach, in contrast, is non-adaptive and does the subset selection only once in the beginning, thereby achieving resource and memory efficiency along with compute-efficiency at training time. Our experiments show that both the variants of our model outperform several methods on the quality of the subset chosen and further demonstrate that our method can be used for choosing the best architecture from a set of architectures.

## 1 INTRODUCTION

In the last decade, deep neural networks have enhanced the performance of the state-of-the-art ML models dramatically. However, these neural networks often demand massive data to train, which renders them heavily contingent on availability of high performance computing machinery, *e.g.*, GPUs, CPUs, RAMs, storage disks, etc. However, such resources entail heavy energy consumption, excessive $CO_2$ emission and maintenance cost.

Driven by this challenge, a recent body of work focus on suitably selecting a subset of instances, so that the model can be quickly trained using lightweight computing infrastructure (Boutsidis et al., 2013; Kirchhoff & Bilmes, 2014; Wei et al., 2014a; Bairi et al., 2015; Liu et al., 2015; Wei et al., 2015; Lucic et al., 2017; Mirzasoleiman et al., 2020b; Kaushal et al., 2019; Killamsetty et al., 2021a;b;c). However, these existing data subset selection algorithm are discrete combinatorial algorithms, which share three key limitations. (1) Scaling up the combinatorial algorithms is often difficult, which imposes significant barrier against achieving efficiency gains as compared to training with entire data. (2) Many of these approaches are adaptive in nature, i.e, the subset changes as the model training progresses. As a result, they require access to the entire training dataset and while they provide compute-efficiency, they do not address memory and resource efficiency challenges of deep model training. (3) The subset selected by the algorithm is tailored to train only a given specific model and it cannot be used to train another model. Therefore, the algorithm cannot be shared across different models. We discuss the related work in detail in Appendix A.

## 1.1 PRESENT WORK

Responding to the above limitations, we develop SUBSELNET, a trainable subset selection framework, which— once trained on a set of model architectures and a dataset— can quickly select a small training subset such that it can be used to train a new (test) model, without a significant drop in accuracy. Our setup is non-adaptive in that it learns to select the subset before the training starts for a new architecture, instead of adaptively selecting the subset during the training process. We initiate our investigation by writing down an instance of combinatorial optimization problem that outputs a subset specifically for one given model architecture. Then, we gradually develop SUBSELNET, by building upon this setup. SUBSELNET comprises of the following novel components.

**Neural model approximator.** The key blocker in scaling up a model-specific combinatorial subset selector across different architectures is the involvement of the model parameters as optimization variables along with the candidate data subset. To circumvent this blocker, we design a neural model approximator which aims to approximate the predictions of a trained model for any given architecture. Thus, such a model approximator can provide per instance accuracy provided by a new (test) model without explicitly training it. This model approximator works in two steps. First, it translates a given model architecture into a set of embedding vectors using graph neural networks (GNNs). Similar to the proposal of Yan et al. (2020) it views a given model architecture as a directed graph between different operations and, then outputs the node embeddings by learning a variational graph autoencoder (VAE) in an unsupervised manner. Due to such nature of the training, these node embeddings represent only the underlying architecture— they do not capture any signal from the predictions of the trained model. Hence, in the next step, we build a neural model encoder which uses these node embeddings and the given instance to approximate the prediction made by the trained model. The model encoder is a transformer based neural network which combines the node embedding using self-attention induced weights to obtain an intermediate graph representation. This intermediate representation finally combines with the instance vector $x$ to provide the prediction of the trained architecture.

**Subset sampler.** Having computed the prediction of a trained architecture, we aim to choose a subset of instances that would minimize the predicted loss and at the same time, offers a good representation of the data. Our subset sampler takes the approximate model output and an instance as input and computes a selection score. Then it builds a logit vector using all these selection scores, feeds it into a multinomial distribution and samples a subset from it. This naturally leads to two variants of the model.

*Transductive-SUBSELNET:* The first variant is transductive in nature. Here, for each new architecture, we utilize the predictions from the model approximator to build a continuous surrogate of the original combinatorial problem and solve it to obtain the underlying selection scores. Thus, we still need to solve a fresh optimization problem for every new architecture. However, the direct predictions from the model approximator allow us to skip explicit model training. This makes this strategy extremely fast both in terms of memory and time. We call this transductive subset selector as Transductive-SUBSELNET.

*Inductive-SUBSELNET:* In contrast to Transductive-SUBSELNET, the second variant does not require to solve any optimization problem. Consequently, it is extremely fast. Instead, it models the scores using a neural network which is trained across different architectures to minimize the entropy regularized sum of the prediction loss. We call this variant as Inductive-SUBSELNET.

We compare our method against six state-of-the-art methods on three real world datasets, which show that Transductive-SUBSELNET (Inductive-SUBSELNET) provides the best (second best) trade off between accuracy and inference time as well as accuracy and memory usage, among all the methods. This is because (1) our subset selection method does not require any training at any stage of subset selection for a new model; and, (2) our approach is non-adaptive and does the subset selection before the training starts. In contrast, most state-of-the-art data subset selection approaches are adaptive, in that the subset selection adapts as the training progresses, and as a result, they require access to the entire data at training time. Finally, we design a hybrid version of the model, where given a budget, we first select a larger set of instances using Inductive-SUBSELNET, and then extract the required number of instances using Transductive-SUBSELNET. We observe that such a hybrid approach allow us to make a smooth transition between the trade off curves from Inductive-SUBSELNET to Transductive-SUBSELNET.

## 2 DEVELOPMENT OF PROPOSED MODEL: SUBSELNET

In this section, we setup the notations and write down the combinatorial subset selection problem for efficient training. This leads us to develop a continuous optimization problem which would allow us to generalize the combinatorial setup across different models.

### 2.1 NOTATIONS

We are given a set of training instances $\{(\boldsymbol{x}_i, y_i)\}_{i \in D}$ where we use $D$ to index the data. Here, $\boldsymbol{x}_i \in \mathbb{R}^{d_x}$ are features and $y_i \in \mathcal{Y}$ as the labels. In our experiments, we consider $\mathcal{Y}$ as a set of categorical labels. However, our framework can also be used for continuous labels. We use $m$ to denote a neural architecture and represent its parameterization as $m_\theta$. We also use $\mathcal{M}$ to denote the set of neural architectures. Given an architecture $m \in \mathcal{M}$, $G_m = (V_m, E_m)$ provides the graph representation of $m$, where the nodes $u \in V_m$ represent the *operations* and the $e = (u_m, v_m)$ indicates an edge, where the output given by the operation represented by the node $u_m$ is fed to one of the operands of the operation given by the node $v_m$. Finally, we use $H(\cdot)$ to denote the entropy of a probability distribution and $\ell(m_\theta(\boldsymbol{x}), y)$ as the cross entropy loss hereafter.

### 2.2 COMBINATORIAL SUBSET SELECTION FOR EFFICIENT LEARNING

We are given a dataset $\{(\boldsymbol{x}_i, y_i)\}_{i \in D}$ and a model architecture $m \in \mathcal{M}$ with its neural parameterization $m_\theta$. The goal of a subset selection algorithm is to select a small subset of instances $S$ with $|S| = n << |D|$ such that, training $m_\theta$ on the subset $S$ gives nearly same accuracy as training on the entire dataset $D$. Existing works (Killamsetty et al., 2021b; Sivasubramanian et al., 2021; Killamsetty et al., 2021a) adopt different strategies to achieve this goal, but all of them aim to simultaneously optimize for the model parameters $\theta$ as well as the candidate subset $S$. At the outset, we may consider the following optimization problem.

$$\underset{\theta, S \subset D : |S| = b}{\text{minimize}} \sum_{i \in S} \ell(m_\theta(\boldsymbol{x}_i), y_i) - \lambda \, \text{DIVERSITY}(S), \tag{1}$$

where $b$ is the budget, $\text{DIVERSITY}(S)$ measures the representativeness of $S$ with respect to the whole dataset $D$ and $\lambda$ is a regularizing coefficient. One can use submodular functions (Fujishige, 2005; Iyer, 2015) like Facility Location, graph cut, or Log-Determinants to model $\text{DIVERSITY}(S)$. Here, $\lambda$ trades off between training loss and diversity. Such an optimization problem indeed provides an optimal subset $S$ that results in high accuracy.

**Bottlenecks of the combinatorial optimization.** The optimization problem (1) imposes the following challenges. (1) It demands explicit training of $m_\theta$ which can be expensive in terms of both memory and time. (2) The training of $m_\theta$ every time for a new architecture $m$ prevents the subset $S$ from being generalizable— one needs to solve the optimization (1) again to find $S$ for an unseen model architecture. We address these challenges by designing a neural surrogate of the objective (1), which would lead to generalization of subset selection across efficient training of different models.

### 2.3 COMPONENTS OF SUBSELNET MODEL

Next, we sketch our proposed model SUBSELNET that leads to substituting the optimization (1) with its neural surrogate. It consists of two key components: (i) neural approximator of the trained model and (ii) the subset sampler. Figure 4 in Appendix B illustrates our model.

**Approximator of the trained model $m_{\theta^*}$.** First, we design a neural network $F_\phi$ which would approximate the predictions of the trained model $m_{\theta^*}$ for different architectures $m \in \mathcal{M}$. Given the dataset $\{(\boldsymbol{x}_i, y_i)_{i \in D}\}$ and a model architecture $m \in \mathcal{M}$, we first feed the underlying DAG $G_m$ into a graph neural network $\text{GNN}_\alpha$ with parameter $\alpha$, which outputs the representations of the nodes of the $G_m$, *i.e.*, $\boldsymbol{H}_m = \{\boldsymbol{h}_u\}_{u \in V_m}$. Next, we feed $\boldsymbol{H}_m$ and the instance $\boldsymbol{x}_i$ into an encoder $g_\beta$

$$F_\phi(G_m, \boldsymbol{x}_i) \approx m_{\theta^*}(\boldsymbol{x}_i) \quad \text{for } m \in \mathcal{M}. \tag{2}$$
$$\text{Here, } F_\phi(G_m, \boldsymbol{x}_i) = g_\beta(\text{GNN}_\alpha(G_m), \boldsymbol{x}_i). \tag{3}$$

Here, $\phi = \{\alpha, \beta\}$, and $\theta^*$ is the set of learned parameters of the model $m_\theta$ on the dataset $D$.

**Subset sampler.** We design a subset sampler using a probabilistic model $\text{Pr}_\pi(\bullet)$. Given a budget $|S| \le b$, it sequentially draws instances $S = \{s_1, ..., s_b\}$ from a softmax distribution of the logit vector $\pi \in \mathbb{R}^{|D|}$ where $\pi(\boldsymbol{x}_i, y_i)$ indicates a score for the element $(\boldsymbol{x}_i, y_i)$. Having chosen the

first $t$ instances $S_t = \{s_1, ..s_t\}$ from $D$, it draws the $(t + 1)$-th element $(\boldsymbol{x}, y)$ from the remaining instances in $D$ with a probability proportional to $\exp(\pi(\boldsymbol{x}, y))$ and then repeat it for $b$ times. Thus, the probability of selecting the ordered set of elements $S = \{s_1, ..., s_b\}$ is given by

$$\Pr{}_\pi(S) = \prod_{t=0}^{b} \frac{\exp(\pi(\boldsymbol{x}_{s_{t+1}}, y_{s_{t+1}}))}{\sum_{\tau \in D \setminus S_t} \exp(\pi(\boldsymbol{x}_{s_\tau}, y_{s_\tau}))} \tag{4}$$

We would like to highlight that we use $S$ as an ordered set of elements, selected in a sequential manner. However, such an order does not affect the trained model which is inherently invariant of permutations of the training data, it only affects the choice of $S$.

**Training objective.** Using the Eqs. (2) and (4), we replace the combinatorial optimization problem in Eq. (1) with a continuous optimization problem, across different model architectures $m \in \mathcal{M}$. To that goal, we define

$$\Lambda(S; m; \pi, F_\phi) = \sum_{i \in S} \ell(F_\phi(G_m, \boldsymbol{x}_i), y_i) - \lambda H(\Pr{}_\pi(\bullet)) \tag{5}$$

$$\underset{\pi, \phi}{\text{minimize}} \sum_{m \in \mathcal{M}} \underset{S \in \Pr{}_\pi(\bullet)}{\mathbb{E}} \left[ \Lambda(S; m; \pi, F_\phi) + \sum_{i \in S} \gamma KL(F_\phi(G_m, \boldsymbol{x}_i), m_{\theta^*}(\boldsymbol{x}_i)) \right] \tag{6}$$

Here, we use entropy on the subset sampler $H(\Pr_\pi(\bullet))$ to model the diversity of samples in the selected subset. We call our neural pipeline, which consists of the model approximator $F_\phi$ and the subset selector $\pi$, as SUBSELNET. In the above, $\gamma$ penalizes the difference between the output of model approximator and the prediction made by the trained model, which allows us to generalize the training of different models $m \in \mathcal{M}$ through the model $F_\phi(G_m, \boldsymbol{x}_i)$.

## 2.4 TRANSDUCTIVE-SUBSELNET AND INDUCTIVE-SUBSELNET MODELS

The optimization (6) suggests that once $F_\phi$ is trained, we can use it to compute the output of the trained model $m_{\theta^*}$ for an unseen architecture $m'$ and use it to compute $\pi$. This already removes a significant overhead of model training and facilitates fast computation of $\pi$. This leads us to develop two types of models based on how we can compute $\pi$, as follows.

**Transductive-SUBSELNET.** The first variant of the model is transductive in terms of computation of $\pi$. Here, once we train the model approximator $F_\phi$, then we compute $\pi$ by solving the optimization problem explicitly with respect to $\pi$, every time when we wish to select data subset for a new architecture. Given a trained model $F_\phi$ and a new model architecture $m' \in \mathcal{M}$, we solve the optimization problem: $\min_\pi \mathbb{E}_{S \in \Pr_\pi(\bullet)}[\Lambda(S; m; \pi, F_\phi)]$ to find the subset sampler $\Pr_\pi$ during inference time for a new architecture $m'$. Such an optimization still consumes time during inference. However, it is still significantly faster than the combinatorial methods (Killamsetty et al., 2021b;a; Mirzasoleiman et al., 2020a; Sivasubramanian et al., 2021) thanks to sidestepping the explicit model training using a model approximator.

**Inductive-SUBSELNET.** In contrast to the transductive model, the inductive model does not require explicit optimization of $\pi$ in the face of a new architecture. To that aim, we approximate $\pi$ using a neural network $\pi_\psi$. This takes two signals as inputs - the dataset $D$ and the outputs of the model approximator for different instances $\{F_\phi(G_m, \boldsymbol{x}_i) \,|\, i \in D\}$, and finally outputs a score for each instance $\pi_\psi(\boldsymbol{x}_i, y_i)$. Under Inductive-SUBSELNET, the optimization (6) becomes:

$$\underset{\psi, \phi}{\text{minimize}} \sum_{m \in \mathcal{M}} \underset{S \in \Pr{}_{\pi_\psi}(\bullet)}{\mathbb{E}} \left[ \Lambda(S; m; \pi_\psi, F_\phi) + \sum_{i \in S} \gamma KL(F_\phi(G_m, \boldsymbol{x}_i), m_{\theta^*}(\boldsymbol{x}_i)) \right] \tag{7}$$

Such an inductive model can select an optimal distribution of the subset that should be used to efficiently train any model $m_\theta$, without explicitly training $\theta$ or searching for the underlying subset.

## 3 NEURAL PARAMETERIZATION OF SUBSELNET

In this section, we describe the neural parametrization of SUBSELNET. SUBSELNET consists of two key components, $F_\phi$ and $\pi_\psi$. Specifically, Transductive-SUBSELNET has only one neural component which is $F_\phi$, whereas, Inductive-SUBSELNET has both $F_\phi$ and $\pi_\psi$.

### 3.1 NEURAL PARAMETERIZATION OF $F_\phi$

The approximator $F_\phi$ consists of two components: (i) a graph neural network $\text{GNN}_\alpha$ which maps $G_m$, the DAG of an architecture, to the node representations $\boldsymbol{H}_m = \{\boldsymbol{h}_u\}_{u \in V_m}$ and (ii) a model encoder

$g_\beta$ which takes $\boldsymbol{H}_m$ and the instance $\boldsymbol{x}_i$ as input and approximates $m_{\theta^*}(\boldsymbol{x}_i)$, *i.e.*, the prediction made by the trained model. Therefore, $F_\phi(G_m, \boldsymbol{x}) = g_\beta(\text{GNN}_\alpha(G_m), \boldsymbol{x}_i)$. Here, $\phi = \{\alpha, \beta\}$.

**Computation of architecture embedding using** $\text{GNN}_\alpha$**.** Given a model $m \in \mathcal{M}$, we compute the representations $\boldsymbol{H}_m = \{\boldsymbol{h}_u | u \in V_m\}$ by using a graph neural network $\text{GNN}_\alpha$ parameterized with $\alpha$, following the proposal of Yan et al. (2020). We first compute the feature vector $\boldsymbol{f}_u$ for each node $u \in V_m$ using the one-hot encoding of the associated *operation* (*e.g.*, max, sum, *etc.*) and then feed it into a neural network to compute an initial node representation, as given below.

$$\boldsymbol{h}_u[0] = \text{INITNODE}_\alpha(\boldsymbol{f}_u) \tag{8}$$

Then, we use a message passing network, which collects signals from the neighborhood of different nodes and recursively compute the node representations (Yan et al., 2020; Xu et al., 2018b; Gilmer et al., 2017). Given a maximum number of recursive layers $K$ and the node $u$, we compute the node embeddings $\boldsymbol{H}_m = \{\boldsymbol{h}_u | u \in V_m\}$ by gathering information from the $k < K$ hops using $K$ recursive layers as follows.

$$
\begin{aligned}
\boldsymbol{h}_{(u,v)}[k-1] &= \text{EDGEEMBED}_\alpha(\boldsymbol{h}_u[k-1], \boldsymbol{h}_v[k-1]) \\
\boldsymbol{h}_u'[k-1] &= \text{SYMMAGGR}_\alpha(\{\boldsymbol{h}_{(u,v)}[k-1] \,|\, v \in \text{Nbr}(u)\}) \\
\boldsymbol{h}_u[k] &= \text{UPDATE}_\alpha(\boldsymbol{h}_u[k-1], \boldsymbol{h}_u'[k-1]).
\end{aligned}
\tag{9}
$$

Here, $\text{Nbr}(u)$ is the set of neighbors of $u$. We use SYMMAGGR as a simple sum aggregator and both UPDATE and EDGEEMBED are injective mappings, as used in (Xu et al., 2018b). Note that trainable parameters from EDGEEMBED, SYMMAGGR and UPDATE are decoupled. They are represented as the set of parameters $\alpha$. Finally, we obtain our node representations as:

$$\boldsymbol{h}_u = [\boldsymbol{h}_u[0], .., \boldsymbol{h}_u[K-1]]. \tag{10}$$

**Model encoder** $g_\beta$**.** Having computed the architecture representation $\{\boldsymbol{h}_u \,|\, u \in V_m\}$, we next design the model encoder which leverages these embeddings to predict the output of the trained model $m_{\theta^*}(\boldsymbol{x}_i)$. To this aim, we developed a model encoder $g_\beta$ parameterized by $\beta$ that takes $\boldsymbol{H}_m$ and $\boldsymbol{x}_i$ as input and attempts to predict $m_{\theta^*}(\boldsymbol{x}_i)$, *i.e.*, $g_\beta(\boldsymbol{H}_m, \boldsymbol{x}_i) \approx m_{\theta^*}(\boldsymbol{x}_i)$. It consists of three steps. In the first step, we generate a permutation invariant order on the nodes. Next, we feed the representations $\{\boldsymbol{h}_u\}$ in this order into a self-attention based transformer layer. Finally, we combine the output of the transformer and the instance $\boldsymbol{x}_i$ using a feedforward network to approximate the model output.

*Node ordering using BFS order.* We first sort the nodes using breadth-first-search (BFS) order $\rho$. Similar to You et al. (2018), this sorting method produces a permutation-invariant sequence of nodes and captures subtleties like skip connections in the network structure $G_m$

*Attention layer.* Given the BFS order $\rho$, we pass the representations $\boldsymbol{H}_m = \{\boldsymbol{h}_u \,|\, u \in V_m\}$ in the sequence $\rho$ through a self-attention based transformer network. Here, the Query, Key and Value functions are realized by matrices $\boldsymbol{W}_{\text{query}}, \boldsymbol{W}_{\text{key}}, \boldsymbol{W}_{\text{value}} \in \mathbb{R}^{\dim(\boldsymbol{h}) \times k}$ where $k$ is a tunable width. Thus, for each node $u \in V_m$, we have:

$$\text{Query}(\boldsymbol{h}_u) = \boldsymbol{W}_{\text{query}}^\top \boldsymbol{h}_u, \quad \text{Key}(\boldsymbol{h}_u) = \boldsymbol{W}_{\text{key}}^\top \boldsymbol{h}_u, \quad \text{Value}(\boldsymbol{h}_u) = \boldsymbol{W}_{\text{value}}^\top \boldsymbol{h}_u \tag{11}$$

Using these quantities, we compute an attention weighted vector $\boldsymbol{\zeta}_u$ given by:

$$\text{Att}_u = \boldsymbol{W}_c^T \sum_v a_{u,v} \text{Value}(\boldsymbol{h}_v) \quad \text{with, } a_{u,v} = \text{SOFTMAX}_v\left(\text{Query}(\boldsymbol{h}_u)^\top \text{Key}(\boldsymbol{h}_v)/\sqrt{k}\right) \tag{12}$$

Here $k$ is the dimension of the latent space, the softmax operation is over the node $v$, and $\boldsymbol{W}_c \in \mathbb{R}^{k \times \dim(\boldsymbol{h})}$. Subsequently, for each node $u$, we use a feedforward network, preceded and succeeded by layer normalization operations, which are given by the following set of equations.

$$\boldsymbol{\zeta}_{u,1} = \text{LN}(\text{Att}_u + \boldsymbol{h}_u; \gamma_1, \gamma_2), \boldsymbol{\zeta}_{u,2} = \boldsymbol{W}_2^\top \text{RELU}(\boldsymbol{W}_1^\top \boldsymbol{\zeta}_{u,1}), \boldsymbol{\zeta}_{u,3} = \text{LN}(\boldsymbol{\zeta}_{u,1} + \boldsymbol{\zeta}_{u,2}; \gamma_3, \gamma_4)$$

Here, LN is the layer normalization operation (Ba et al., 2016). Finally, we feed the vector $\zeta_{u,3}$ for the last node $u$ in the sequence $\rho$, *i.e.*, $u = \rho(|V_m|)$ along with the feature vector $\boldsymbol{x}_i$ into a feed-forward network parameterized by $\boldsymbol{W}_F$ to model the prediction $m_{\theta^*}(\boldsymbol{x}_i)$. Thus, the final output of the model encoder $g_\beta(\boldsymbol{H}_m, \boldsymbol{x}_i)$ is given by

$$\boldsymbol{o}_{m,\boldsymbol{x}_i} = \text{FF}_{\beta_2}(\boldsymbol{\zeta}_{\rho_{|V_m|},3}, \boldsymbol{x}_i) \tag{13}$$

Here, $\boldsymbol{W}_\bullet$ and $\gamma_\bullet$ are trainable parameters and collectively form the set of parameters $\beta$.

### 3.2 NEURAL ARCHITECTURE OF INDUCTIVE-SUBSELNET

We approximate $\pi$ using a neural network $\pi_\psi$ using a neural network which takes three inputs – $(\boldsymbol{x}_j, y_j)$, the corresponding output of the model approximator, *i.e.*, $\boldsymbol{o}_{m,\boldsymbol{x}_j} = F_\phi(G_m, \boldsymbol{x}_j)$ and the node representation matrix $\boldsymbol{H}_m$ and provides us a positive selection score $\pi_\psi(\boldsymbol{H}_m, \boldsymbol{x}_j, y_j, \boldsymbol{o}_{m,\boldsymbol{x}_j})$. In practice, $\pi_\psi$ is a three-layer feed-forward network, which contains Leaky-ReLU activation functions for the first two layers and sigmoid activation at the last layer.

## 4 PARAMETER ESTIMATION AND INFERENCE

Given a dataset $\{(\boldsymbol{x}_i, y_i) \mid i \in D\}$ and the output of the trained models $\{m_{\theta^*}(\boldsymbol{x}_i)\}_{i \in D}$, our goal is to estimate $\phi$ and $\pi$ (resp. $\psi$) for the transductive (inductive) model. We first illustrate the bottlenecks that prevent us from end-to-end training for estimating these parameters. Then, we introduce a multi-stage training method to overcome these limitations. Finally, we present the inference method.

### 4.1 BOTTLENECK FOR END TO END TRAINING

End to end optimization of the above problem is difficult for the following reasons. (i) Our architecture representation $\boldsymbol{H}_m$ only represents the architectures and thus should be independent of parameter of the architecture $\theta$ and the instances $\boldsymbol{x}$. End to end training can make them sensitive to these quantities. (ii) To enable the model approximator $F_\phi$ accurately fit the output of the trained model $m_\theta$, we need an explicit training for $\phi$ with the target $m_\theta$. Adding the corresponding loss as an additional regularizer imposes an additional hyperparameter tuning.

### 4.2 MULTI-STAGE TRAINING

In our multi-stage training method, we first train the model approximator $F_\phi$ by minimizing the sum of the KL divergence between the gold output probabilities, and then train our subset sampler $\Pr_\pi$ (resp. $\Pr_{\pi_\psi}$) for the transductive (inductive) model as well as fine-tuning $\phi$.

**Training the model approximator $F_\phi$.** We train $F_\phi$ in two steps. In the first step, we perform unsupervised training of $\text{GNN}_\alpha$ using graph variational autoencoder (GVAE). This ensures that the architecture representations $\boldsymbol{H}_m$ remain insensitive to the model parameters. We build the encoder and decoder of our GVAE by following existing works on graph VAEs (Yan et al., 2020) in the context graph based modeling of neural architectures. Given a graph $G_m$, the encoder $q(\mathcal{Z}_m \mid G_m)$ which takes the node embeddings $\{\boldsymbol{h}_u\}_{u \in V_m}$ and maps it into the latent space $\mathcal{Z}_m = \{\boldsymbol{z}_u\}_{u \in V_m}$. Specifically, we model the encoder $q(\mathcal{Z}_m \mid G_m)$ as: $q(\boldsymbol{z}_u \mid G_m) = \mathcal{N}(\mu(\boldsymbol{h}_u), \Sigma(\boldsymbol{h}_u))$. Here, both $\mu$ and $\Sigma$ are neural networks. Given a latent representation $\mathcal{Z}_m = \{\boldsymbol{z}_u\}_{u \in V_m}$, the decoder models a generative distribution of the graph $G_m$ where the presence of an edge is modeled as Bernoulli distribution $\text{BERNOULLI}(\sigma(\boldsymbol{z}_u^\top \boldsymbol{z}_v))$. Thus, we model the decoder as:

$$p(G_m \mid \mathcal{Z}) = \prod_{(u,v) \in E_m} \sigma(\boldsymbol{z}_u^\top \boldsymbol{z}_v) \cdot \prod_{(u,v) \notin E_m} [1 - \sigma(\boldsymbol{z}_u^\top \boldsymbol{z}_v)] \tag{14}$$

Here, $\sigma$ is a parameterized sigmoid function. Finally, we estimate $\alpha, \mu, \Sigma$ and $\sigma$ by maximizing the evidence lower bound (ELBO) as follows:

$$\max_{\alpha, \mu, \Sigma, \sigma} \mathbb{E}_{\mathcal{Z} \sim q(\bullet \mid G_m)}[p(G_m \mid \mathcal{Z})] - \text{KL}(q(\bullet \mid G_m) || \text{prior}(\bullet)) \tag{15}$$

Next, we train our model encoder $g_\beta$ by minimizing the KL-Divergence between the approximated prediction $g_\beta(\boldsymbol{H}_m, \boldsymbol{x}_i)$ and the ground truth prediction $m_{\theta^*}(\boldsymbol{x}_i)$, where both these quantities are probabilities across different classes. Hence, the training problem is as follows:

$$\underset{\beta}{\text{minimize}} \sum_{i \in D, m \in \mathcal{M}} \text{KL}(m_{\theta^*}(\boldsymbol{x}_i) || g_\beta(\boldsymbol{H}_m, \boldsymbol{x}_i)) \tag{16}$$

**Training of the subset sampler.** Finally, we fine-tune $g_\beta$ and train $\pi$ by solving (6) for the Transductive-SUBSELNET (likewise train $\pi_\psi$ by solving (7) for Inductive-SUBSELNET).

### 4.3 INFERENCE

During inference, our goal is to select a subset $S$ with $|S| = b$ for a new model $m'$, which would facilitate efficient training of $m'$. As discussed in Section 2.4, we compute $\pi$ for Transductive-SUBSELNET by explicitly solving the optimization problem: $\min_\pi \mathbb{E}_{S \in \Pr_\pi(\bullet)}[\Lambda(S; m; \pi, F_\phi)]$ and then draw $S \sim \Pr_\pi(\bullet)$. For Inductive-SUBSELNET, we draw $S \sim \Pr_{\pi_{\hat{\psi}}}(\bullet)$ where $\hat{\psi}$ is the learned value of $\psi$ during training.

### 4.4 OVERVIEW OF TRAINING AND INFERENCE ROUTINES

Algorithms 1 and 2 summarize the algorithms for the training and inference procedure.

**Training Subroutines.** The training phase for both, Transductive-SUBSELNET first utilizes the TRAINAPPROX routine to train the model approximator given the dataset, trained model parameters, and the set of neural architectures. Internally, the routine calls the TRAINGNN subroutine to train the parameters ($\alpha$) of the GNN network, BFSORDERING subroutine to reorder the embeddings based on the BFS order and the TRAINMODELENC subroutine to train the attention-based model encoder's parameters ($\beta$). The TRAININDUCTIVE routine further calls the TRAINPI subroutine to train the parameters of the neural subset selector.

**Inference Subroutines.** Given an unseen architecture and parameters of the trained neural networks, the inference phase for both variants of SUBSELNET first generates the model encoder output for all the data points. Post this, the INFERTRANSDUCTIVE routine solves the optimization problem on $\pi$ explicitly for the unseen architecture and selects the subset from the dataset. On the other hand, INFERINDUCTIVE utilizes the trained parameters of the neural subset selector. Finally, both routines call the TRAINNEWMODEL to train and evaluate the unseen architecture on selected subset.

---

**Algorithm 1** Training procedure

1: **function** TRAINTRANSDUCTIVE($D, \mathcal{M}, \{\theta^*\}$)
2:    $\hat{\alpha}, \hat{\beta}, \boldsymbol{H}_m \leftarrow$ TRAINAPPROX($D, \mathcal{M}, \{\theta^*\}$)

---

1: **function** TRAININDUCTIVE($D, \mathcal{M}, \{\theta^*\}$)
2:    $\hat{\alpha}, \hat{\beta}, \boldsymbol{H}_m \leftarrow$ TRAINAPPROX($D, \mathcal{M}, \{\theta^*\}$)
3:    $\boldsymbol{o} \leftarrow [g_{\hat{\beta}}(\{\boldsymbol{H}_m, \boldsymbol{x}_i\})]_{i,m}$
4:    $\hat{\psi} \leftarrow$ TRAINPI($\boldsymbol{o}, \{\boldsymbol{H}_m\}, \{\boldsymbol{x}_i\}$)

---

1: **function** TRAINAPPROX($D, \mathcal{M}, \{\theta^*\}$)
2:    $\hat{\alpha} \leftarrow$ TRAINGNN($\mathcal{M}$)
3:    **for** $m \in \mathcal{M}^{\text{train}}$ **do**
4:       $\boldsymbol{H}_m \leftarrow$ GNN$_{\hat{\alpha}}(m)$
5:       POS $\leftarrow$ BFSORDERING($G_m$)
6:    $\hat{\beta} \leftarrow$ TRAINMODELENC($\{\boldsymbol{x}_i\}$, POS, $\{\theta^*\}$)

---

**Algorithm 2** Inference procedure

1: **function** INFERTRANSDUCTIVE($D, \hat{\alpha}, \hat{\beta}, m'$)
2:    $\boldsymbol{H}_{m'} \leftarrow$ GNN$_{\hat{\alpha}}(m')$
3:    $F_\phi(G_{m'}, \boldsymbol{x}_i) \leftarrow g_{\hat{\beta}}(\boldsymbol{H}_{m'}, \boldsymbol{x}_i) \ \forall i \in D$
4:    $\pi^* \leftarrow \min_\pi \mathbb{E}_{S \in \text{Pr}_\pi(\bullet)}[\Lambda(S; m'; \pi; F_\phi)]$
5:    $S^* \sim \text{Pr}_{\pi^*}(\bullet)$
6:    TRAINNEWMODEL($m'; S^*$)

---

1: **function** INFERINDUCTIVE($D, \hat{\alpha}, \hat{\beta}, m'$)
2:    $\boldsymbol{H}_{m'} \leftarrow$ GNN$_{\hat{\alpha}}(m')$
3:    $F_\phi(G_{m'}, \boldsymbol{x}_i) \leftarrow g_{\hat{\beta}}(\boldsymbol{H}_{m'}, \boldsymbol{x}_i) \ \forall i \in D$
4:    Compute $\pi_{\hat{\psi}}(\boldsymbol{x}_i, y_i) \ \forall i \in D$
5:    $S^* \sim \text{Pr}_{\pi_{\hat{\psi}}}(\bullet)$
6:    TRAINNEWMODEL($m'; S^*$)

---

## 5 EXPERIMENTS

In this section, we provide comprehensive evaluation of SUBSELNET against several strong baselines on three real world datasets. In Appendix D, we present additional results.

### 5.1 EXPERIMENTAL SETUP

**Datasets.** We use FMNIST (Xiao et al., 2017), CIFAR10 (Krizhevsky et al., 2014) and CIFAR100 (Krizhevsky et al., 2009) datasets for our experiments. We transform an input image $\boldsymbol{X}_i$ to a vector $\boldsymbol{x}_i$ of dimension $2048$ by feeding it to a pre-trained ResNet50 v1.5 (**?**) model and using the output from the penultimate layer as the image representation.

**Model architectures and baselines.** We use model architectures from NAS-Bench-101 (Ying et al., 2019) for our experiments. We compare Transductive-SUBSELNET and Inductive-SUBSELNET against two non-adaptive subset selection methods – (i) Facility location (Fujishige, 2005; Iyer, 2015) where we maximize $FL(S) = \sum_{j \in D} \max_{i \in S} \boldsymbol{x}_i^\top \boldsymbol{x}_j$ to find $S$, (ii) Pruning (Sorscher et al., 2022), and four adaptive subset selection methods – (iii) Glister (Killamsetty et al., 2021b), (iv) Grad–Match (Killamsetty et al., 2021a), (v) EL2N (Paul et al., 2021), (vi) GraNd (Paul et al., 2021); and; (vii) Full selection where we use complete training data. The non-adaptive subset selectors select the subset before the training begins and thus, never access the rest of the training set again during the training iterations. On the other hand, the adaptive subset selectors refine the choice of subset during training iterations and thus they need to access the full training set at each training iteration. Appendix C contains additional details about the baselines.

**Evaluation protocol.** We split the model architectures $\mathcal{M}$ into 60% training ($\mathcal{M}_{\text{tr}}$), 20% validation ($\mathcal{M}_{\text{val}}$) and 20% test ($\mathcal{M}_{\text{test}}$) folds. Similarly, we split the dataset $D$ into $D_{\text{tr}}$, $D_{\text{val}}$ and $D_{\text{test}}$. We present $\mathcal{M}_{\text{tr}}$, $\mathcal{M}_{\text{val}}$, $D_{\text{tr}}$ and $D_{\text{val}}$ to our method and estimate $\hat{\phi}$ and $\hat{\psi}$ (for Inductive-SUBSELNET

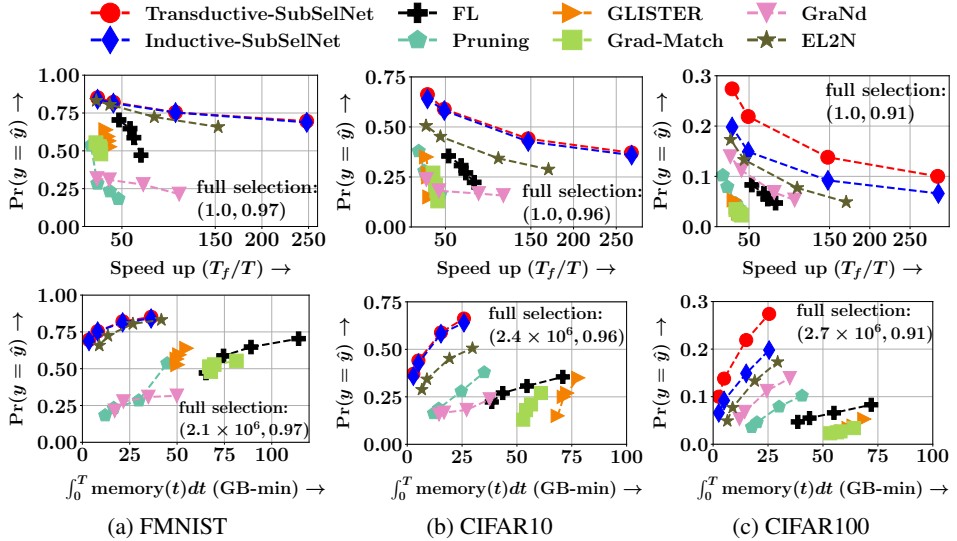

Figure 1: Trade off between accuracy and speedup (top row) and accuracy and memory consumption (bottom row) for all the methods – Facility location (Fujishige, 2005; Iyer, 2015), Pruning (Sorscher et al., 2022), Glister (Killamsetty et al., 2021b), Grad-Match (Killamsetty et al., 2021a), EL2N (Paul et al., 2021); GraNd (Paul et al., 2021); and; Full selection on all three datasets - FMNIST, CIFAR10 and CIFAR100. In all cases, we vary $|S| = b \in (0.005|D|, 0.05|D|)$ and measure accuracy on 20% test architectures and 20% test instances.

model). None of the baseline methods supports any generalizable learning protocol across different model architectures and thus cannot leverage the training architectures during test.

Given an architecture $m' \in \mathcal{M}_{\text{test}}$, we select the subset $S$ from $D_{\text{tr}}$ using our subset sampler ($\Pr_\pi$ for Transductive-SUBSELNET or $\Pr_{\pi_{\hat{\psi}}}$ for Inductive-SUBSELNET). Similarly, all the non-adaptive subset selectors select $S \subset D_{\text{tr}}$ using their own algorithms. Once $S$ is selected, we train the test models $m' \in \mathcal{M}_{\text{test}}$ on $S$. We perform our experiments with different $|S| = b \in (0.005|D|, 0.05|D|)$ and compare the performance between different methods using three quantities: (1) Accuracy $\Pr(y = \hat{y})$ measured using $\frac{1}{|D_{\text{test}}|}\sum_{i \in D_{\text{test}}}\sum_{m' \in \mathcal{M}_{\text{test}}}\mathbf{1}(\max_j m'_{\theta*}(\boldsymbol{x}_i)[j] = y_i)$. (2) Computational efficiency, *i.e.*, the speedup achieved with respect to training with full dataset. It is measured with respect to $T_f/T$. Here, $T_f$ is the time taken for training with full dataset; and, $T$ is the time taken for the entire inference task, which is the average time for selecting subsets across the test models $m' \in \mathcal{M}_{\text{test}}$ plus the average training time of these test models on the respective selected subsets. (3) Resource efficiency in terms of the amount of memory consumed during the entire inference task, described in item (2), which is measured as $\int_0^T \text{memory}(t)\,dt$ where $\text{memory}(t)$ is amount of memory consumed at timestamp $t$.

## 5.2 RESULTS

**Comparison with baselines.** Here, we compare different methods in terms of the trade off between accuracy and computational efficiency as well as accuracy and resource efficiency. In Figure 1, we probe the variation between these quantities by varying the size of the selected subset $|S| = b \in (0.005|D|, 0.05|D|)$. We make the following observations. (1) Our methods trade-off between accuracy vs. computational efficiency as well as accuracy vs. resource efficiency more effectively than all the methods. For FMNIST, both the variants of our method strikingly output 75% accuracy, whereas they are 100 times faster than full selection. Transductive-SUBSELNET performs slightly better than Inductive-SUBSELNET in terms of the overall trade-off between accuracy and efficiency for FMNIST and CIFAR10 datasets. However, for CIFAR100, Transductive-SUBSELNET performs significantly better than Inductive-SUBSELNET. The time taken for both Transductive-SUBSELNET and Inductive-SUBSELNET seems comparable— this is because the subset selection time for both of them are significantly less than the final training time on the selected subset. (2) EL2N is the second best method. It provides the best trade-off between accuracy and time as well as accuracy and GPU memory, among all the baselines. It aims at choosing difficult training instances having high prediction error. As a result, once trained on them, the model can predict the labels of easy instances

too. However, it chooses instances after running the initial few epochs. (3) FL adopts a greedy algorithm for subset selection and therefore, it consumes a large time and memory during subset selection itself. Consequently, the overall efficiency significantly decreases although the complexity of the training time on the selected subset remains the same as our models in terms of time and memory. (4) In addition to EL2N, Glister, Grad-Match and GraNd are adaptive subset selection methods that operate with moderately small ($> 5\%$) subset sizes. In a region, where the subset size is extremely small, *i.e.*, $1\% - 5\%$, they perform very poorly. Moreover, they maximize a monotone function at each gradient update step, which results in significant overhead in terms of time. These methods process the entire training data to refine the choice of the subset and consequently, they end up consuming a lot of memory. (5) GraNd selects the instances having high uncertainty after running each model for five epochs and often the model is not well trained by then.

**Finer analysis of the inference time.** Next, we demarcate the subset selection phase from the training phase of the test models on the selected subset during the inference time analysis. Table 2 summarizes the results for top three non-adaptive subset selection methods for $b = 0.005|D|$ on CIFAR100. We observe that: (1)

Table 2: Inference time in seconds

|  | Trans. | Induct. | FL |
|---|---|---|---|
| Subset selection | 0.23 | 0.067 | 226.29 |
| Training | 70.1 | 70.1 | 70.1 |

the final training times of all three methods are roughly same; (2) the selection time for Transductive-SUBSELNET is significantly more than Inductive-SUBSELNET, although it remains extremely small as compared to the final training on the inferred subset; and, (3) the selection time of FL is large— as close as 323% of the training time.

**Hybrid-SUBSELNET.** From Figure 1, we observe that Transductive-SUBSELNET performs significantly better than Inductive-SUBSELNET. However, since Transductive-SUBSELNET solves a fresh optimization problem for each new architecture, it performs better at the cost of time and GPU memory. On the other hand, Inductive-SUBSELNET performs significantly worse as it relies on a trained neural network to learn the same optimization problem. Here, we design a hybrid version of our model, called as Hybrid-SUBSELNET. Here, given the budget of the subset $b$, we first choose $B > b$ instances using Inductive-SUBSELNET and the final $b$ instances by running the explicit optimization routines in Transductive-SUBSELNET. Figure 3 sum-

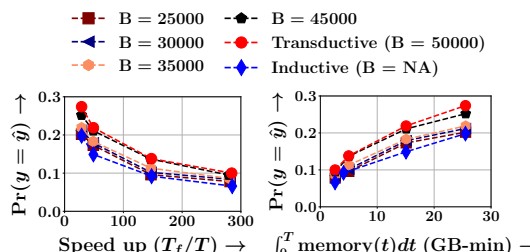

Figure 3: Hybrid-SUBSELNET

marizes the results for $B = \{25K, 30K, 35K, 45K, 50K\}$. We observe that the trade off curves for the Hybrid-SUBSELNET lie in between Inductive-SUBSELNET and Transductive-SUBSELNET. For low value of $B$, *i.e.*, $B = 25K$, the trade off line of Hybrid-SUBSELNET remains close to Inductive-SUBSELNET. As we increase $B$, the trade-off curve of accuracy vs speed up as well as the accuracy vs GPU usage becomes better, which allows Hybrid-SUBSELNET to smoothly transition from the trade off curve of Inductive-SUBSELNET to Transductive-SUBSELNET. At $B = 45K$, the trade-off curve almost coincides with Transductive-SUBSELNET. Such properties allow a user to choose an appropriate $B$ that can accurately correspond to a target operating point in the form of (Accuracy, Speed up) or (Accuracy, memory usage).

## 6 CONCLUSION

In this work, we develop SUBSELNET, a subset selection framework, which can be trained on a set of model architectures, to be able to predict a suitable training subset before training a model, for an unseen architecture. To do so, we first design a neural model approximator, which predicts the output of a new candidate architecture without explicitly training it. We use that output to design transductive and inductive variants of our model. The transductive model solves a small optimization problem to compute the subset for a new architecture $m$ every single time. In contrast, the inductive model resorts to a neural subset sampler instead of an optimizer.

Our work does not incorporate the gradients of the trained model in model approximator and it would be interesting to explore its impact on the subset selection. Further we can extend our setup to an adaptive setting, where we can incorporate signals from different epochs with a sequence encoder to train a subset selector.

## 7 ETHICS STATEMENT

We do not foresee any negative impact of our work from ethics viewpoint.

## 8 REPRODUCIBILITY STATEMENT

We uploaded the code in supplementary material. Details of implementation are given in Appendix C.

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

# Efficient Data Subset Selection to Generalize Training Across Models: Transductive and Inductive Networks (Appendix)

## A  RELATED WORK

Our work is closely related to representation learning for model architectures, network architecture search, data subset selection.

**Representation learning for model architectures.** Recent work in network representation learning use GNN based encoder-decoder to encapsulate the local structural information of a neural network into a fixed-length latent space (Zhang et al., 2019; Ning et al., 2020; Yan et al., 2020; Lukasik et al., 2021). By employing an asynchronous message passing scheme over the directed acyclic graph (DAG), GNN-based methods model the propagation of input data over the actual network structure. Apart from encodings based solely on the structure of the network, White et al. (2020); Yan et al. (2021) produce computation-aware encodings that map architectures with similar performance to the same region in the latent space. Following the work of Yan et al. (2020), we use a graph isomorphism network as an encoder but instead of producing a single graph embedding, our method produces a collection of node embeddings, ordered by breadth-first-search (BFS) ordering of the nodes. Our work also differs in that we do not employ network embeddings to perform downstream search strategies. Instead, architecture embeddings are used in training a novel *model approximator* that predicts the logits of a particular architecture, given an architecture embedding and a data embedding.

**Network architecture search.** There is an ever-increasing demand for the automatic search of neural networks for various tasks. The networks discovered by NAS methods often come from an underlying search space, usually designed to constrain the search space size. One such method is to use cell-based search spaces (Luo et al., 2018; Zoph et al., 2017; Liu et al., 2017; Pham et al., 2018; Ying et al., 2019; Dong & Yang, 2020). Although we utilize the NAS-Bench-101 search space for architecture retrieval, our work is fundamentally different from NAS. In contrast to the NAS methods, which search for the best possible architecture from the search space using either sampling or gradient-descent based methods (Baker et al., 2017; Zoph & Le, 2016; Real et al., 2017; 2018; Liu et al., 2018; Tan et al., 2018), our work focuses on efficient data subset selection given a dataset and an architecture, which is sampled from a search space. Our work utilizes graph representation learning on the architectures sampled from the mentioned search spaces to project an architecture under consideration to a continuous latent space, utilize the model expression from the latent space as proxies for the actual model and proceed with data subset selection using the generated embedding, model proxy and given dataset.

**Data subset selection.** Data subset selection is widely used in literature for efficient learning, coreset selection, human centric learning, *etc.* Several works cast the efficient data subset selection task as instance of submodular or approximate-submodular optimization problem (Killamsetty et al., 2021a; Wei et al., 2014a;b;c; Killamsetty et al., 2021b; Sivasubramanian et al., 2021). Another line of work focus on selecting coresets which are expressed as the weighted combination of subset of data, approximating some characteristics, *e.g.*, loss function, model prediction (Feldman, 2020; Mirzasoleiman et al., 2020b; Har-Peled & Mazumdar, 2004; Boutsidis et al., 2013; Lucic et al., 2017). Our work is closely connected to simultaneous model learning and subset selection (De et al., 2021; 2020; Sivasubramanian et al., 2021). These existing works focus on jointly optimizing the training loss, with respect to the subset of instances and the parameters of the underlying model. Among them (De et al., 2021; 2020) focus on distributing decisions between human and machines, whereas (Sivasubramanian et al., 2021) aims for efficient learning. However, these methods adopt a combinatorial approach for selecting subsets and consequently, they are not generalizable across architectures. In contrast, our work focuses on differentiable subset selection mechanism, which can generalize across architectures.

# B ILLUSTRATION OF SUBSELNET

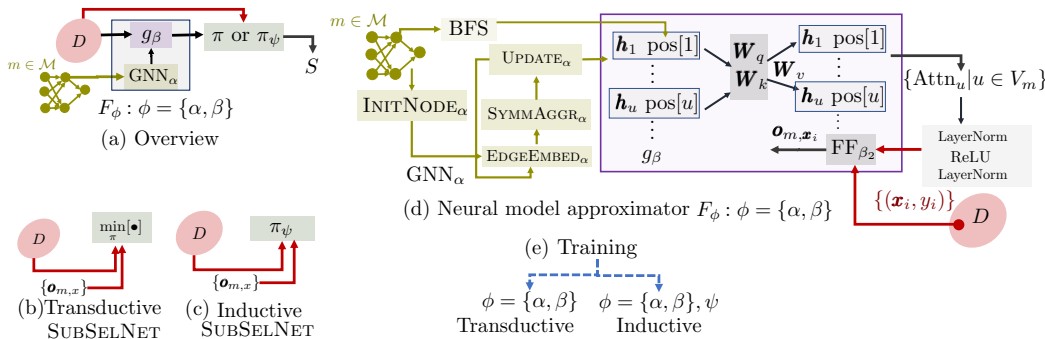

Figure 4: Illustration of SUBSELNET. (a) Overview of SUBSELNET: Given a model architecture $m \in \mathcal{M}$, SUBSELNET takes the graph structure of model architecture $G_m$ and a dataset $D$ as input to a neural model approximator $F_\phi$, which predicts the output of the trained model $m_{\theta^*}(\boldsymbol{x})$. Then this is fed as input to the subset sampler $\pi$ (for transductive model) or $\pi_\psi$ (for inductive model) to obtain the training subset $S$. $F_\phi$ consists of a graph neural network GNN$_\alpha$ and a model encode $g_\beta$. (b) Transductive-SUBSELNET: Given the dataset $D$ and the predictions $\{\boldsymbol{o}_{m,\boldsymbol{x}_j} \,|\, j \in D\}$ from the model approximator $F_\phi$, we compute the training subset $S$ by solving an optimization problem for each $m \in \mathcal{M}$. (c) Inductive-SUBSELNET: Here, we do not have to solve the optimization problem each time for a new architecture $m$. Instead, we use a neural network $\pi_\psi$ which learns to solve the optimization problem of the transductive model during training and thus, directly provides the training subset $S$ during test. (d) Architecture of the neural model approximator $F_\phi$: It first feeds $G_m$ into a graph neural network GNN$_\alpha$, which outputs the node representations $\{\boldsymbol{h}_u\}$. Together with a sequence of positional encoding obtained from a BFS ordering, $\{\boldsymbol{h}_u\}$ are fed into a attention network followed by a feed forward network, which outputs $\{\boldsymbol{o}_{m,\boldsymbol{x}_i}\}$ for an instance $(\boldsymbol{x}_i, y_i)$. (e) The output of training procedure for the two variants of our model.

## C  ADDITIONAL DETAILS ABOUT EXPERIMENTAL SETUP

### C.1  DATASET

**Datasets** ($D$).

Table 5: A brief description of the datasets used along with the transformations applied during training

| Dataset | No. of Classes | Train-Test Split | Shape | Transformations Applied |
|---------|---------------|------------------|-------|------------------------|
| CIFAR10 | 10 | (50K,10K) | 32x32x3 | RandomCrop, Normalize |
| CIFAR100 | 100 | (50K,10K) | 32x32x3 | RandomCrop, Normalize |
| FMNIST | 10 | (60K,10K) | 28x28x1 | Normalize |

**Architectures** ($\mathcal{M}$). Although our task is not Neural Architecture Search, we leverage the NASBench-101 search space as an architecture pool. The cell-based search space was designed for the benchmarking of various NAS methods. It consists of $423,624$ unique architectures with the following constraints – (1) number of nodes in each cell is at most 7, (2) number of edges in each cell is at most 9, (3) barring the input and output, there are three unique operations, namely $1 \times 1$ convolution, $3 \times 3$ convolution and $3 \times 3$ max-pool. We utilize the architectures from the search space in generating the sequence of embeddings along with sampling architectures for the training and testing of the encoder and datasets for the subset selector.

### C.2  IMPLEMENTATION DETAILS ABOUT BASELINES

**Facility Location (FL).** We implemented facility location on all the three datasets using the apricot [1] library. The similarity matrix was computed using Euclidean distance between data points, and the objective function was maximized using the naive greedy algorithm.

**Pruning.** It selects a subset from the entire dataset based on the uncertainty of the datapoints while partial training. In our setup, we considered ResNet-18 as a master model, which is trained on each dataset for 5 epochs. Post training, the uncertainty measure is calculated based on the probabilities of each class, and the points with highest uncertainty are considered in the subset. We train the master model at a learning rate of 0.025.

**Glister and Grad-Match.** We implemented GLISTER (Killamsetty et al., 2021b) and Grad-Match (Killamsetty et al., 2021a) using the CORDS library. We trained the models for 50 epochs, using batch size of 20, and selected the subset after every 10 epochs. The loss was minimized using SGD with learning rate of 0.01, momentum of 0.9 and weight decay with regularization constant of $5 \times 10^{-4}$. We used cosine annealing for scheduling the learning rate with $T_{max}$ of 50 epochs, and used 10% of the training data as the validation set. Details of specific hyperparameters for stated as follows.

*Glister* uses a greedy selection approach to minimize a bi-level objective function. In our implementation, we used stochastic greedy optimization with learning rate 0.01, applied on the data points of each mini-batch. Online-Glister approximates the objective function with a Taylor series expansion up to an arbitrary number of terms to speed up the process; we used 15 terms in our experiments.

*Grad-Match* applies the orthogonal matching (OMP) pursuit algorithm to the data points of each mini-batch to match gradient of a subset to the entire training/validation set. Here, we set the learning rate is set to 0.01. The regularization constant in OMP is 1.0 and the algorithm optimizes the objective function within an error margin of $10^{-4}$.

**GraNd.** This is an adaptive subset selection strategy in which the norm of the gradient of the loss function is used as a score to rank a data point. The gradient scores are computed after the model has trained on the full dataset for the first few epochs. For the rest of epochs, the model is trained only on the top-$k$ data points, selected using the gradient scores. In our implementation, we let the model train on the full dataset for the first 5 epochs, and computed the gradient of the loss only with respect to the last layer fully connected layer.

**EL2N.** When the loss function used to compute the GraNd scores is the cross entropy loss, the norm of the gradient for a data point $\mathbf{x}$ can be approximated by $\mathbb{E}||p(\mathbf{x}) - y||_2$, where $p(\mathbf{x})$ is the discrete

---

[1] https://github.com/jmschrei/apricot

probability distribution over the classes, computed by taking `softmax` of the logits, and $y$ is the one-hot encoded true label corresponding to the data point **x**. Similar to our implementation of GraNd, we computed the EL2N scores after letting the models train on the full data for the first 5 epochs.

### C.3 IMPLEMENTATION DETAILS ABOUT OUR MODEL

**GNN$_\alpha$.** As we utilize NASBench-101 space as the underlying set of neural architectures, each computational node in the architecture can comprise of one of five *operations* and the one-hot-encoded feature vector $\mathbf{f}_u$. Since the set is cell-based, there is an injective mapping between the neural architecture and the cell structure. We aim to produce a sequence of embeddings for the cell, which in turn corresponds to that of the architecture. For each architecture, we use the initial feature $\mathbf{f}_u \in R^5$ in (8) as a five dimensional one-hot encoding for each operation. This is fed into INITNODE (8) to obtain an 16 dimensional output. Here, INITNODE consists of a $5 \times 16$ linear, ReLU and $16 \times 16$ linear layers cascaded with each other. Each of EDGEEMBED and UPDATE consists of a $5 \times 128$ linear-BatchNorm-ReLU cascaded with a $128 \times 16$ linear layer. Moreover, the symmetric aggregator is a sum aggregator.

We repeat this layer $K$ times, and each iteration gathers information from $k < K$ hops. After all the iterations, we generate an embedding for each node, and following (You et al., 2018) we use the BFS-tree based node-ordering scheme to generate the sequence of embeddings for each network.

The GVAE-based architecture was trained for 10 epochs with the number of recursive layers $K$ set to 5, and the Adam optimizer was used with learning rate of $10^{-3}$. The entire search space was considered as the dataset, and a batch-size of 32 was used. Post training, we call the node embeddings collectively as the architecture representation.

To train the latent space embeddings, the parameters $\alpha$ are trained in an encoder-decoder fashion using a variational autoencoder. The mean $\mu$ and variance $\sigma$ on the final node embeddings $\boldsymbol{h}_u$ are:

$$\mu = \text{FCN}\left(\left[\boldsymbol{h}_u\right]_{u \in V_m}\right) \text{ and } \sigma = \exp\left(\text{FCN}\left(\left[\boldsymbol{h}_u\right]_{u \in V_m}\right)\right)$$

The decoder aims to reconstruct the original cell structure (i.e the nodes and the corresponding operations), which are one-hot encoded. It is modeled using single-layer fully connected networks followed by a sigmoid layer.

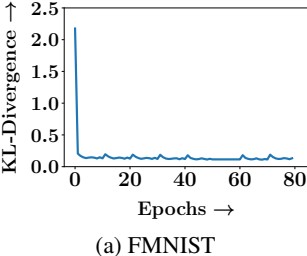 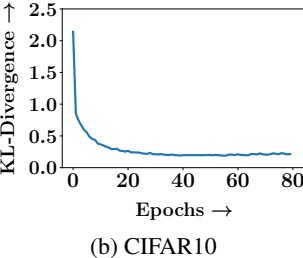

(a) FMNIST          (b) CIFAR10

Figure 6: Kullback-Leibler divergence values $(\text{KL}(m_{\theta*}(\boldsymbol{x}_i) \| g_\beta(\boldsymbol{H}_m, \boldsymbol{x}_i)))$ computed during the training of the model encoder $g_\beta$ over 80 epochs.

**Model Encoder $g_\beta$.** The model encoder $g_\beta$ is essentially a single-head attention block that acts on a sequence of node embeddings $\boldsymbol{H}_m = \{h_u | u \in V_m\}$. The Query, Key and Value matrices, $\boldsymbol{W}_{\text{query}}$, $\boldsymbol{W}_{\text{key}}$ and $\boldsymbol{W}_{\text{value}} \in \mathbb{R}^{16 \times 8}$, and the matrix $\boldsymbol{W}_C \in \mathbb{R}^{8 \times 16}$. The fully connected network acting on $\zeta_{u,1}$ consists of matrices $W_1 \in \mathbb{R}^{16 \times 64}$ and $W_2 \in \mathbb{R}^{64 \times 16}$. All the trainable matrices along with the layer normalizations were implemented using the `Linear` and `LayerNorm` functions in Pytorch. The last item of the output sequence $\zeta_{u,3}$ is concatenated with the data embedding $\boldsymbol{x}_i$ and fed to another 2-layer fully-connected network with hidden dimension 256 and dropout probability of $0.3$. The model encoder is trained by minimizing the KL-divergence between $g_\beta(\boldsymbol{H}_m, \boldsymbol{x}_i)$ and $m_{\theta*}(\boldsymbol{x}_i)$. We used an AdamW optimizer with learning rate of $10^{-3}$, $\epsilon = 10^{-8}$, `betas` $= (0.9, 0.999)$ and weight decay of $0.005$. We also used Cosine Annealing to decay the learning rate, and used gradient clipping with maximum norm set to 5. Figure 6 shows the convergence of the outputs of the model encoder $g_\beta(\boldsymbol{H}_m, \boldsymbol{x}_i)$ with the outputs of the model $m_{\theta*}(\boldsymbol{x}_i)$.

**Neural Network $\pi_\psi$.** The inductive model is a three-layer fully-connected neural network with two Leaky ReLU activations and a sigmoid activation after the last layer. The input to $\pi_\psi$ is the concatenation $(\boldsymbol{H}_m; \boldsymbol{o}_{m,i}; \boldsymbol{x}_i; y_i)$. The hidden dimensions of the two intermediary layers are $64$ and $16$, and the final layer is a single neuron that outputs the score corresponding to a data point $\boldsymbol{x}_i$. While training $\pi_\psi$ we add a regularization term $\lambda'(\sum_{i \in D} \pi_\psi(\boldsymbol{H}_m, \boldsymbol{o}_{m,i}, \boldsymbol{x}_i, y_i) - |S|)$ to ensure that nearly $|S|$ samples have high scores out of the entire dataset $D$. Both the regularization constants $\lambda$ (in equation 6) and $\lambda'$ are set to $0.1$. We train the model weights using an Adam optimizer with a learning rate of $0.001$. During training, at each iteration we draw instances using $\mathrm{Pr}_\pi$ and use the log-derivative trick to compute the gradient of the objective. During each computation step, we use one instance of the ranked list to compute the unbiased estimate of the objective in (6) .

# D  ADDITIONAL EXPERIMENTS

## D.1  ABLATION STUDY

We perform ablation study of SUBSELNET from three perspectives.

**Impact of ablation of subset sampler.** First, we attempt to understand the impact of the subset sampler. To that aim, we compare the performance of SUBSELNET against two baselines, namely - Bottom-$b$-loss and Bottom-$b$-loss+gumbel. In Bottom-$b$-loss, we sort the data instances based on their predicted loss $\ell(F_\phi(G_m, \boldsymbol{x}), y)$ and consider those points with the bottom $b$ values. In Bottom-$b$-loss+gumbel, we add noise sampled from the gumbel distribution with $\mu = 0$ and $\beta = 0.025$, and sort the instances based on these noisy loss values, *i.e.*, $\ell(F_\phi(G_m, \boldsymbol{x}), y) + \mathrm{Gumbel}(0, \beta = 0.025)$.

We observe that Bottom-$b$-loss and Bottom-$b$-loss+gumbel do not perform that well in spite of being efficient in terms of time and memory. Figure 7 compares the performance of the variants of SUBSELNET, Bottom-$b$-loss and Bottom-$b$-loss+gumbel.

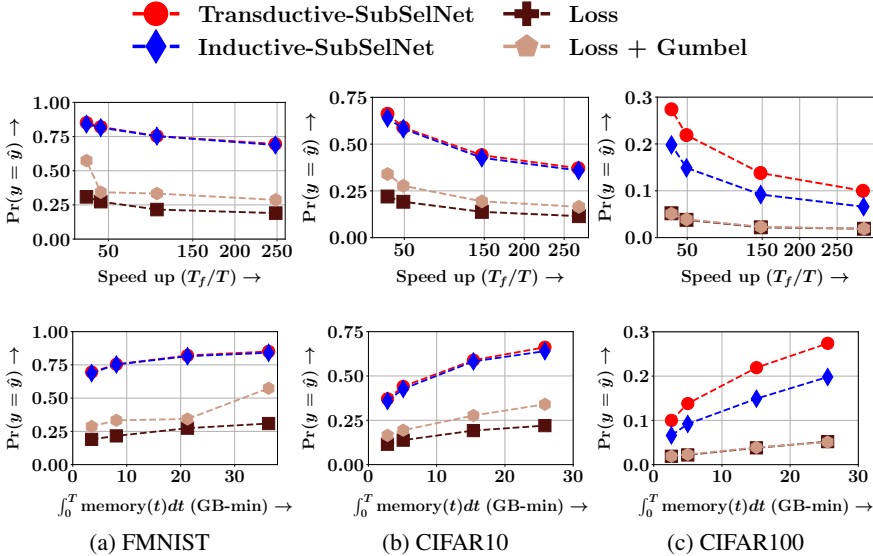

Figure 7: Comparison of Transductive-SUBSELNET and Inductive-SUBSELNET with Bottom-$b$-loss (Loss) and Bottom-$b$-loss+gumbel (Loss + Gumbel). In the former, we select top-$b$ instances in terms of their predicted loss $\ell(F_\phi(G_m, \boldsymbol{x}), y)$ computed using the model approximator In Bottom-$b$-loss+gumbel, we add gumbel noise $\mathrm{Gumbel}(0, 0.025)$ to the loss and sort the instances based on these noisy loss values.

**Exploring alternative architecture of the model encoder $g_\beta$.** We consider three alternative architecture to our current model encoder $g_\beta$.

- FEEDFORWARD: We consider a two-layer fully-connected network, in which we concatenate the mean of $\boldsymbol{H}_m$ with $\boldsymbol{x}_i$. We used ReLu activation between the layers and the hidden dimension was set to 256. We used dropout for regularization with probability 0.3.

- DEEPSET: We consider permutation invariant networks of the form $\rho(\sum_{h \in H} \phi(h); \boldsymbol{x}_i)$ where $\rho$ and $\phi$ are neural networks and $H$ is the sequence under consideration. We $\rho$ is a fully-connected network with 4 layers, ReLU activation, and hidden dimension of 64, and $\phi$ is a two-layer fully-connected network with ReLU activation and has output dimension 10.
- LSTM: We consider an LSTM-based encoder with hidden dimension of 16 and dropout probability of 0.2. The output of the last LSTM block is concatenated with $\boldsymbol{x}_i$ and fed to a linear layer with hidden dimension 256, dropout probability of 0.3 and ReLU as the activation function.

Since the goal of the model encoder is to produce outputs which mimic the architectures, we measure the KL divergence between the outputs of the gold models and of the encoder to denote the closeness of the output distribution. Table. 8 summarizes performance of different model encoders. We make the following observations: (1) Transformer-based model encoder outperforms every other method by a significant margin across both the datasets. (2) The BFS sequential modeling of an architecture with transformers leads to better representation that enables closer model approximation compared to other sequential methods like LSTM. (3) Non-sequential model approximators like Feedforward and DeepSets led to poor model approximation.

Table 8: Comparison of the performance of several model encoder architectures $g_\beta$ on the CIFAR-10 and FMNIST datasets, based on the Kullback–Leibler divergence values between the gold model outputs and predicted model outputs.

| Model approximator | CIFAR-10 | FMNIST |
|---|---|---|
| Feedforward | 0.171 | 0.124 |
| DeepSet | 0.105 | 0.122 |
| LSTM | 0.102 | 0.113 |
| Self-Attention Based | **0.089** | **0.109** |

**Performance of subset selectors using different model encoders.** We consider three different design choices of model approximator (our (Transformer), Feedforward, and LSTM) along with three different subset selection strategies (Our subset sampler, top-b instances based on uncertainty, and top-b based on loss) which result in nine different combinations of model approximation and subset selection strategies. We measure uncertainty using the entropy of the predicted distribution of the target classes and report the average test accuracy of the models when they are trained on the underlying pre-selected subset in the following table -

Table 9: Test accuracy of the nine combinations of model approximators and selection strategies on the pre-selected CIFAR10 subset of size 5%.

| Design | Accuracy |
|---|---|
| Feedforward + SUBSELNET | 0.527 |
| Feedforward + Uncertainty | 0.329 |
| Feedforward + Loss | 0.296 |
| LSTM + SUBSELNET | 0.526 |
| LSTM + Uncertainty | 0.417 |
| LSTM + Loss | 0.283 |
| Transformer + SUBSELNET | 0.548 |
| Transformer + Uncertainty | 0.198 |
| Transformer + Loss | 0.210 |

We make the following observations -

1. The complete design of our method, i.e., Our model approximator (Transformer) + Our subset sampler (SUBSELNET) performs best.
2. If we use simple unsupervised subset selection heuristics, e.g., loss or uncertainty based subset selection, then our model approximator performs much worse than Feedforward or

LSTM, whereas this trend is opposite if we use our subset sampler for selecting the subset. This may be due to overfitting of the transformer architecture in presence of uncertainty or loss based selection, which is compensated by our subset sampler.

## D.2 RECOMMENDING MODEL ARCHITECTURE

When dealing with a pool of architectures designed for the same task, choosing the correct architecture for the task might be a daunting task - since it is impractical to train all the architectures from scratch. In view of this problem, we show that training on smaller carefully chosen subsets might be beneficial for a quicker alternative to choosing the correct architectures. We first extract the top $15$ best performing architectures $\mathcal{A}^*$ having highest accuracy, when trained on full data. We mark them as "gold". Then, we gather top $15$ architectures $\mathcal{A}$ when trained on the subset provided by our models. Then, we compare $\mathcal{A}$ and $\mathcal{A}^*$ using the Kendall tau rank correlation coefficient (KTau) along with Jaccard coefficient $|\mathcal{A} \cap \mathcal{A}^*|/|\mathcal{A} \cup \mathcal{A}^*|$.

Figure 10 summarizes the results for top three non-adaptive subset selectors in terms of the accuracy, namely - Transductive-SUBSELNET, Inductive-SUBSELNET and FL. We make the following observations: (1) One of our variant outperforms FL in most of the cases in CIFAR10 and CIFAR100. (2) There is no consistent winner between Transductive-SUBSELNET and Inductive-SUBSELNET, although Inductive-SUBSELNET outperforms both Transductive-SUBSELNET and FL consistently in CIFAR100 in terms of the Jaccard coefficient.

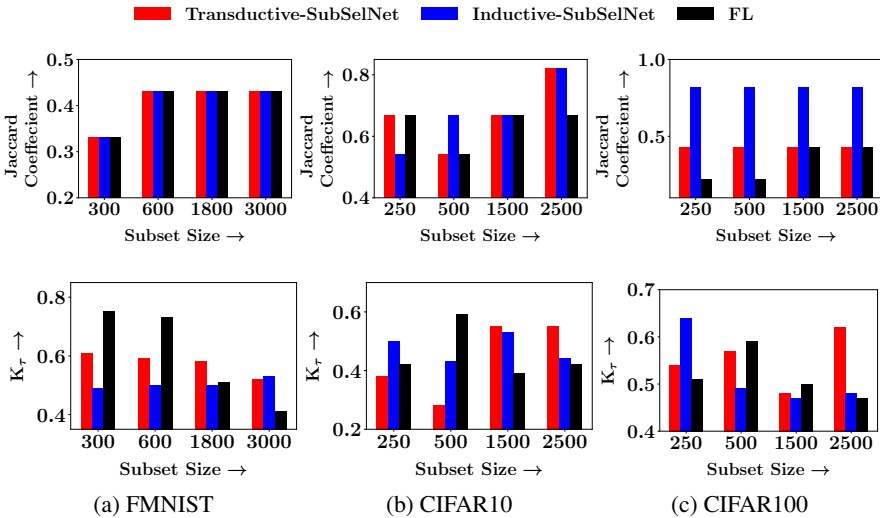

(a) FMNIST         (b) CIFAR10         (c) CIFAR100

Figure 10: Comparison of the top three non-adaptive subset selectors (Transductive-SUBSELNET, Inductive-SUBSELNETand FL) on ranking and choosing of the top-15 architectures on the basis of Jaccard Coefficient and Kendall tau rank correlation coefficient ($K_\tau$).

## D.3 AVOIDING UNDERFITTING AND OVERFITTING

Since the amount of training data is small, there is a possibility of overfitting. However, the coefficient $\lambda$ of the entropy regularizer $\lambda H(Pr_\pi)$, can be increased to draw instances from the different regions of the feature space, which in turn can reduce the overfitting. In practice, we tuned $\lambda$ on the validation set to control such overfitting.

We present the accuracies on (training, validation, test) folds for both Transductive-SUBSELNET and Inductive-SUBSELNET in Table 11.

We make the following observations:

1. From training to test, in most cases, the decrease in accuracy is $\sim 7\%$.
2. This small accuracy gap is further reduced from validation to test. Here, in most cases, the decrease in accuracy is $\sim 4\%$.

We perform early stopping using the validation set which acts as an additional regularizer and therefore, the amount of overfitting is significantly low.

Table 11: Variation of accuracy with subset size of both the variants of SUBSELNET on training, validation and test set of CIFAR10

| Subset Size | Training | | Validation | | Testing | |
|---|---|---|---|---|---|---|
| | Transductive | Inductive | Transductive | Inductive | Transductive | Inductive |
| 10% | 0.728 | 0.660 | 0.702 | 0.632 | 0.678 | 0.606 |
| 20% | 0.852 | 0.673 | 0.809 | 0.658 | 0.770 | 0.644 |
| 40% | 0.890 | 0.691 | 0.856 | 0.678 | 0.825 | 0.666 |
| 70% | 0.942 | 0.738 | 0.912 | 0.717 | 0.884 | 0.698 |

## D.4 PERFORMANCE OF SUBSET SELECTION STRATEGIES ON LARGER SUBSET SIZES

We conducted similar experiments as Section 5.1 for CIFAR10 and FMNIST on larger subset sizes ($b$) of $0.1|D|, 0.2|D|, 0.4|D|$ and $0.7|D|$. For each dataset and the above mentioned subset sizes, we evaluate the decrease in accuracy (ratio of the accuracy on the subset to accuracy on the full dataset), speed-up (ratio of the time taken to train the full dataset to the sum of times taken for subset selection and subset training), and GPU usage in GB-min. We report the variation of these metrics with respect to the subset sizes in the following tables –

Table 12: Decrease in accuracy (Accuracy on selected subset/Accuracy on full data) for CIFAR10 and FMNIST for $b \in (0.1|D|, 0.7|D|)$.

| Subset Size | Dataset | Transductive | Inductive | FacLoc | Pruning | GLISTER | GradMatch | GraNd | EL2N |
|---|---|---|---|---|---|---|---|---|---|
| 10% | | 0.70 | 0.69 | 0.56 | 0.40 | 0.78 | 0.72 | 0.28 | 0.70 |
| 20% | CIFAR10 | 0.81 | 0.74 | 0.67 | 0.61 | 0.88 | 0.87 | 0.30 | 0.79 |
| 40% | | 0.86 | 0.80 | 0.78 | 0.78 | 0.93 | 0.93 | 0.39 | 0.85 |
| 70% | | 0.93 | 0.87 | 0.85 | 0.88 | 0.96 | 0.96 | 0.71 | 0.91 |
| 10% | | 0.92 | 0.90 | 0.86 | 0.42 | 0.95 | 0.95 | 0.35 | 0.91 |
| 20% | FMNIST | 0.94 | 0.91 | 0.90 | 0.62 | 0.96 | 0.96 | 0.40 | 0.93 |
| 40% | | 0.95 | 0.92 | 0.93 | 0.73 | 0.97 | 0.97 | 0.50 | 0.95 |
| 70% | | 0.96 | 0.93 | 0.95 | 0.88 | 0.97 | 0.97 | 0.79 | 0.96 |

Table 13: Speed-up (Time for full training/(Time taken for subset selection + Training on the selected subset)) for CIFAR10 and FMNIST for $b \in (0.1|D|, 0.7|D|)$.

| Subset Size | Dataset | Transductive | Inductive | FacLoc | Pruning | GLISTER | GradMatch | GraNd | EL2N |
|---|---|---|---|---|---|---|---|---|---|
| 10% | | 24.13 | 24.13 | 5.89 | 16.81 | 6.15 | 7.23 | 17.76 | 17.76 |
| 20% | CIFAR10 | 11.82 | 11.83 | 4.70 | 9.74 | 4.19 | 4.91 | 10.06 | 10.06 |
| 40% | | 5.75 | 5.75 | 3.31 | 5.21 | 2.69 | 3.20 | 5.29 | 5.29 |
| 70% | | 3.43 | 3.43 | 2.38 | 3.23 | 1.72 | 2.07 | 3.26 | 3.26 |
| 10% | | 25.68 | 25.67 | 4.32 | 15.60 | 7.70 | 7.77 | 12.28 | 12.28 |
| 20% | FMNIST | 10.57 | 10.57 | 3.48 | 8.35 | 5.42 | 5.47 | 7.29 | 7.29 |
| 40% | | 6.04 | 6.04 | 2.79 | 5.24 | 3.30 | 3.04 | 4.81 | 4.81 |
| 70% | | 3.38 | 3.38 | 2.05 | 3.12 | 1.96 | 2.30 | 2.96 | 2.96 |

Table 14: GPU memory (GB-min) for CIFAR10 and FMNIST for $b \in (0.1|D|, 0.7|D|)$

| Subset Size | Dataset | Transductive | Inductive | FacLoc | Pruning | GLISTER | GradMatch | GraNd | EL2N |
|---|---|---|---|---|---|---|---|---|---|
| 10% | CIFAR10 | 32.66 | 32.65 | 461.77 | 41.41 | 370.08 | 314.87 | 67.80 | 107.70 |
| 20% | | 66.65 | 66.64 | 495.75 | 74.88 | 543.75 | 463.62 | 101.26 | 190.19 |
| 40% | | 137.15 | 137.13 | 566.26 | 144.31 | 847.87 | 711.67 | 170.69 | 361.32 |
| 70% | | 229.97 | 229.95 | 659.07 | 235.73 | 1326.46 | 1101.66 | 262.11 | 586.65 |
| 10% | FMNIST | 23.73 | 23.73 | 641.65 | 32.21 | 168.70 | 167.32 | 88.69 | 87.85 |
| 20% | | 57.64 | 57.64 | 675.56 | 66.12 | 239.84 | 237.38 | 122.60 | 121.76 |
| 40% | | 100.85 | 100.85 | 718.77 | 109.32 | 394.35 | 427.87 | 165.80 | 164.97 |
| 70% | | 180.03 | 180.04 | 797.95 | 188.50 | 664.70 | 565.88 | 244.98 | 244.15 |

Note that in the case of CIFAR10, we denote the decrease factors of 0.91-0.96 in green, and the decrease factors of 0.85 - 0.88 in purple. In case of FMNIST, we denote the decrease factors of 0.94-0.97 in green and the decrease factors of 0.90 - 0.93 in purple.
We make the following observations:

1. We show a better trade-off between accuracy and time and accuracy and memory than almost all the baselines.
2. *Observations in CIFAR10:* When we tuned the subset sizes, we notice that SUBSELNET, GLISTER, Grad-Match and EL2N can achieve a comparable decrease factor of 0.91-0.93. In terms of speed-up and memory usage, we see that
   (a) SUBSELNET achieves a 1.3x speed-up as compared to GLISTER and 1.1x speed-up as compared to Grad-Match and EL2N
   (b) GLISTER consumes 3.7x GPU memory, Grad-Match consumes 3.1x GPU memory and EL2N consumes 2.5x GPU memory as compared to SUBSELNET
   We notice that none of the other subset selection strategies achieve a high-enough accuracy, and we beat them in terms of speed-up and memory usage. Moreover, for the case when the subset selection methods achieve a decrease factor of 0.85 - 0.88, we see that
   (a) SUBSELNET achieves a 2.4x speed-up as compared to FacLoc, 1.8x speed-up as compared to Pruning, 1.4x speed-up as compared to GLISTER, 1.2x speed-up as compared to Grad-Match and 1.1x speed-up as compared to EL2N
   (b) FacLoc consumes 4.8x GPU memory, Pruning consumes 1.7x GPU memory, GLISTER consumes 4x GPU memory, Grad-Match consumes 3.4x GPU memory and EL2N consumes 2.6x GPU memory as compared to SUBSELNET.
3. *Observations in FMNIST:* When we tuned the subset sizes, we notice that SUBSELNET, Facloc, GLISTER, Grad-Match and EL2N can achieve a comparable decrease factor of 0.94-0.97. In terms of speed-up and memory usage, we see that
   (a) SUBSELNET achieves a 3.8x speed-up as compared to FacLoc, 1.4x speed-up as compared to GLISTER and Grad-Match, and 2.2x speed-up as compared to EL2N.
   (b) FacLoc consumes 12.5x GPU Memory, and GLISTER, Grad-Match and EL2N consume 2.9x GPU memory as compared to SUBSELNET.
   We notice that none of the other subset selection strategies achieve a high-enough accuracy, and we beat them in terms of speed-up and memory usage. Moreover, for the case when the subset selection methods achieve a decrease factor of 0.90-0.93, we see that
   (a) SUBSELNET achieves a 7.4x speed-up as compared to FacLoc, 2.1x speed-up as compared to GLISTER, 2.9x speed-up as compared to Grad-Match and 2.1x speed-up as compared to EL2N
   (b) FacLoc consumes 28.5x GPU memory, GLISTER consumes 4.5x GPU memory, Grad-Match consumes 6.1x GPU memory and EL2N consumes 3.7x GPU memory as compared to SUBSELNET.

We present the trade-off between the accuracy and speed-up, and accuracy and memory consumption in Figure 15.

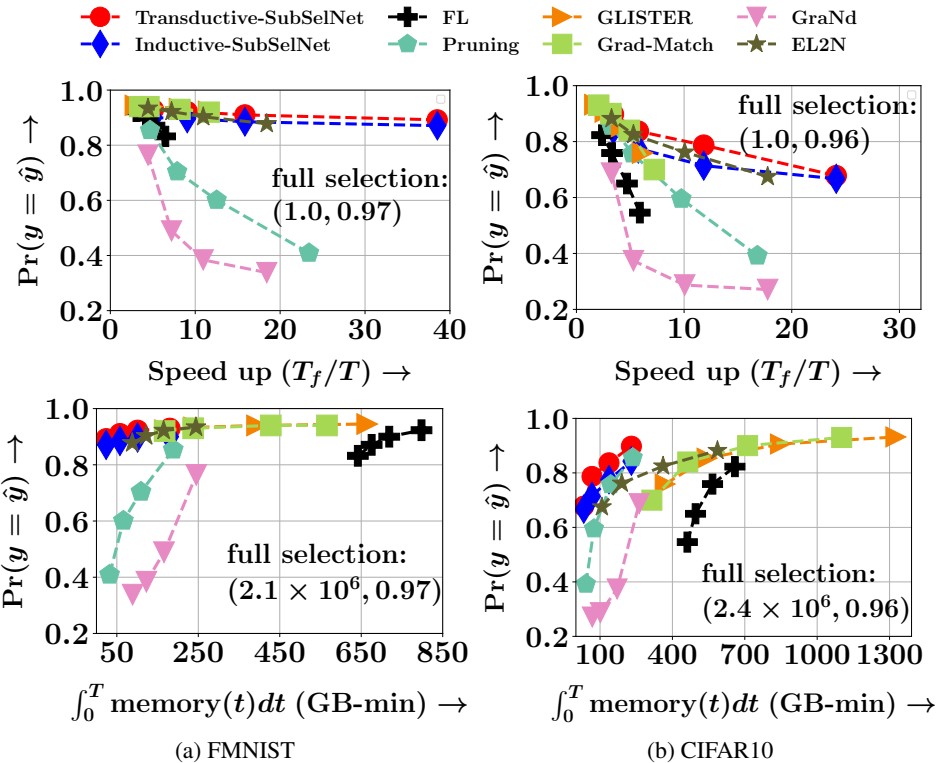

Figure 15: Trade off between accuracy and speedup (top row) and accuracy and memory consumption (bottom row) for all the methods – Facility location (Fujishige, 2005; Iyer, 2015), Pruning (Sorscher et al., 2022), Glister (Killamsetty et al., 2021b), Grad-Match (Killamsetty et al., 2021a), EL2N (Paul et al., 2021); GraNd (Paul et al., 2021); and; Full selection on FMNIST and CIFAR10. In all cases, we vary $|S| = b \in (0.1|D|, 0.7|D|)$.

## E  PROS AND CONS OF USING GNNS

We have used a GNN in our model encoder to encode the architecture representations into an embedding. We chose a GNN for the task due to following reasons -

1. Message passing between the nodes (which may be the input, output, or any of the operations) allows us to generate embeddings that capture the contextual structural information of the node, i.e., the embedding of each node captures not only the operation for that node but also the operations preceding that node to a large extent.
2. It has been shown by (Morris et al., 2019) and (Xu et al., 2018a) that GNNs are as powerful as the Weisfeiler-Lehman algorithm and thus give a powerful representation for the graph. Thus, we obtain smooth embeddings of the nodes/edges that can effectively distill information from its neighborhood without significant compression.
3. GNNs embed model architecture into representations independent of the underlying dataset and the model parameters. This is because it operates on only the nodes and edges— the structure of the architecture and does not use the parameter values or input data.

However, the GNN faces the following drawbacks -

1. GNN uses a symmetric aggregator for message passing over node neighbors to ensure that the representation of any node should be invariant to a permutation of its neighbors. Such a symmetric aggregator renders it a low-pass filter, as shown in (NT & Maehara, 2019), which attenuates important high-frequency signals.
2. We are training one GNN using several architectures. This can lead to the insensitivity of the embedding to change in the architecture. In the context of model architecture, if we change the operation of one node in the architecture (either remove, add or change the operation), then the model's output can significantly change. However, the embedding of GNN may become immune to such changes, since the GNN is being trained over many architectures.

## F    CHOICE OF SUBMODULAR FUNCTION FOR THE OPTIMIZATION PROBLEM

In ( 1) we introduced the original combinatorial problem for subset selection where optimization variable $S$— the subset of instances — makes the underlying problem combinatorial. Here, we can use submodular functions like Graph-Cut, Facility-Location, and Log-Determinant as the diversity functions, which would allow us to use greedy algorithms to maximize the objective in ( 1). But, as discussed in Section 4.1, this suffers from two bottlenecks — expensive computation issues and lack of generalizability. Therefore, we do not follow these approaches and resort to our proposed approach called SUBSELNET.

In contrast to the optimization problem in (1), which was a combinatorial set optimization problem, the optimization problem in SUBSELNET(6) is a continuous optimization problem where the goal is to estimate $Pr_\pi$. In such a problem, where the probability distribution is the key optimization variable, entropy is a more natural measure of diversity than the other submodular measures.

