# OpenReview forum: "Efficient Data Subset Selection to Generalize Training Across Models: Transductive and Inductive Networks"
_ICLR.cc/2023/Conference — Submitted to ICLR 2023_

### Official Review · Reviewer_a8HM · 2022-10-24

**Confidence:** 4
**Correctness:** 2
**Technical Novelty And Significance:** 2
**Empirical Novelty And Significance:** 2
**Recommendation:** 6

**Clarity, Quality, Novelty And Reproducibility:**

Clarity: Clarity is fine. Figures are generally easy to read. Most descriptions are clear.

Quality:  Experiments setups can be improved ( See Weakness 1). Writing can be polished with noticeable grammar mistakes.   For example, Section 5.1 Model architectures and baselines: four non-adaptive  -> four adaptive. Figure1 Caption ;and; -> , and

Novelty:  Neural model approximator using GNN seems new for data selection.


**Strength And Weaknesses:**

Strength:

(1)	Well-motivated problem. Data selection for an unseen architecture is an important problem with real-world impact.

Weakness:

(1)	The main problem of this work is the experiment. The authors choose 0.5% and 5% of training data, however, with such a low subsample rate, we notice a dramatic accuracy drop in Figure 1.  It is hard to judge the effectiveness of any methods with such a significant accuracy drop. The experiment should reflect at what speed up, the model can maintain the same accuracy. Essentially, focusing on the left side of Figure 1. The authors may consider a larger range of sampling rates (5%, 10%, 20%,30%, 40%, 50%, 60%, 70%), following the setup of previous work such as GraNd and Gister.

Question:

(1)	When calculating inference time, are you assume selection and training are conducted in sequential? Is it possible to perform selection in parallel with training when loading a batch of data?

(2)	What is the training cost of training the model approximator? Have the authors evaluated the approximator error?



**Summary Of The Paper:**

This paper introduces a SUBSELNET to select a subset of training data. SUBSELNET is a non-adaptive method as it is agnostic to model architecture and training stage and. The authors design a neural model approximator to approximate the output of any given architecture. Two variants of subset sampler are proposed.  SUBSELNET is compared against state-of-the-art methods on three datasets to demonstrate the tradeoff advantage between accuracy and speed up.

**Summary Of The Review:**

I would recommend weak rejection because of the experiment results. It is hard to judge the effectiveness of the proposed method when all methods exhibit significant accuracy drops.  I would increase my score if the authors could demonstrate the efficiency advantage of the proposed method at no or negligible accuracy loss.

---

> ### Author Response · Authors · 2022-11-18
> **Response to Reviewer a8HM (Part-1/2)**
>
> We would like to thank the reviewer for the comments which would improve our paper.
>
> > *The authors may consider a larger range of sampling rates (5%, 10%, 20%,30%, 40%, 50%, 60%, 70%), following the setup of previous work such as GraNd and Gister*
>
> We increased the subset sizes for CIFAR10 and FMNIST during the rebuttal, and the variation of accuracy, speed-up, and GPU usage(GB-min) are shown in the following tables:
>
> **Table 1: CIFAR10 (Subset size vs decrease factor in accuracy (Accuracy on selected subset/Accuracy on full data). Higher is better):**
> Subset Size (%)| Transductive | Inductive | FacLoc | Pruning | GLISTER | GradMatch | GraNd | EL2N
> -| - | - | - | - | - | - | - | -
> 10| 0.70| 0.69| 0.56| 0.40 | 0.78| 0.72| 0.28|0.70
> 20| 0.81| 0.74| 0.67| 0.61 | $\textcolor{Plum}{0.88}$ | $\textcolor{Plum}{0.87}$| 0.30| 0.79
> 40|$\textcolor{Plum}{0.86}$|0.80| 0.78| 0.78 | $\textcolor{ForestGreen}{0.93}$ | $\textcolor{ForestGreen}{0.93}$| 0.39|$\textcolor{Plum}{0.85}$
> 70| $\textcolor{ForestGreen}{0.93}$| $\textcolor{Plum}{0.87}$| $\textcolor{Plum}{0.85}$| $\textcolor{Plum}{0.88}$| $\textcolor{ForestGreen}{0.96}$ | $\textcolor{ForestGreen}{0.96}$| 0.71|$\textcolor{ForestGreen}{0.91}$
>
> **Table 2: CIFAR10 (Subset size vs speed-up (Time for full training/(Time taken for subset selection + Training on the selected subset)). Higher is better):**
> Subset Size (%) | Transductive | Inductive | FacLoc | Pruning | GLISTER | GradMatch | GraNd | EL2N
> -| - | - | - | - | - | - | - | -
> 10|24.13 |24.13 |5.89 | 16.81 |6.15 | 7.23| 17.76 | 17.76|
> 20|11.82 |11.83 |4.70 | 9.74|$\textcolor{Plum}{4.19}$ | $\textcolor{Plum}{4.91}$ | 10.06 | 10.06 |
> 40|$\textcolor{Plum}{5.75}$|5.75|3.31 | 5.21|$\textcolor{ForestGreen}{2.69}$| $\textcolor{ForestGreen}{3.20}$ |5.29| $\textcolor{Plum}{5.29}$|
> 70|$\textcolor{ForestGreen}{3.43}$|$\textcolor{Plum}{3.43}$|$\textcolor{Plum}{2.38}$ | $\textcolor{Plum}{3.23}$|$\textcolor{ForestGreen}{1.72}$ | $\textcolor{ForestGreen}{2.07}$ | 3.26| $\textcolor{ForestGreen}{3.26}$|
>
> **Table 3: CIFAR10 (Subset size vs GPU memory (GB-min). Lower is better):**
> Subset Size (%)|Transductive |Inductive |FacLoc|Pruning|GLISTER|GradMatch|GraNd |EL2N
> -| - | - | - | - | - | - | - | -
> 10|32.66 |32.65 |461.77|41.41|370.08 |314.87|67.80 |107.70
> 20|66.65 |66.64 |495.75|74.88|$\textcolor{Plum}{543.75}$ |463.62|101.26|190.19
> 40|$\textcolor{Plum}{137.15}$|137.13|566.25|144.31|$\textcolor{ForestGreen}{847.87}$ |$\textcolor{ForestGreen}{711.67}$|170.69|$\textcolor{Plum}{361.32}$
> 70|$\textcolor{ForestGreen}{229.97}$|$\textcolor{Plum}{229.95}$|$\textcolor{Plum}{659.07}$|$\textcolor{Plum}{235.73}$|$\textcolor{ForestGreen}{1326.46}$|$\textcolor{ForestGreen}{1101.66}$|262.11|$\textcolor{ForestGreen}{586.65}$
>
> *In the above tables, for the convenience of comparison, we color the entries that have the decrease factor of accuracy in 0.91-0.96 in $\textcolor{ForestGreen}{\text{green}}$ and those in the range of 0.85-0.88 in $\textcolor{Plum}{\text{purple}}$.*  Hence, the entries of table 1 with same color have similar accuracies. Thus,  the corresponding  entries in the tables with speed up (table 2) and memory (table 3) are marked with same color and only those entries having same color will be compared.
>
> We make the following observations:
>
> (1) We show a better trade-off between accuracy and time and accuracy and memory than almost all the baselines.
>
> (2) *Observations in CIFAR10*: When we tuned the subset sizes, we notice that our method (SubSelNet), GLISTER, Grad-Match, and EL2N can achieve a comparable decrease factor of 0.91-0.93. In terms of speed-up and memory usage, we see that
> SubSelNet achieves a 1.3x speed-up as compared to GLISTER and a 1.1x speed-up as compared to Grad-Match and EL2N.
> GLISTER consumes 3.7x GPU memory, Grad-Match consumes 3.1x GPU memory, and EL2N consumes 2.5x GPU memory as compared to SubSelNet.
>
> We notice that none of the other subset selection strategies achieve a high-enough accuracy, and we beat them in terms of speed-up and memory usage.
>
> Moreover, for the case for all those subset selection methods which achieve a decrease factor of 0.85 - 0.88, we see that –
> SubSelNet achieves a 2.4x speed-up as compared to FacLoc, 1.8x speed-up as compared to Pruning, 1.4x speed-up as compared to GLISTER, 1.2x speed-up as compared to Grad-Match, and 1.1x speed-up as compared to EL2N.
> FacLoc consumes 4.8x GPU memory, Pruning consumes 1.7x GPU memory, GLISTER consumes 4x GPU memory, Grad-Match consumes 3.4x GPU memory, and EL2N consumes 2.6x GPU memory as compared to SubSelNet.
>
>
> (3) We observed similar results in FMNIST. Please refer to Appendix D.4 for details.
>
>
> > *When calculating inference time, do you assume selection and training are conducted in sequential?*
>
> Yes, we conducted the selection and training in a sequential manner, and the inference time is calculated the same way.

---

> > ### Author Response · Authors · 2022-11-18
> > **Response to Reviewer a8HM (Part-2/2)**
> >
> > > *Is it possible to perform selection in parallel with training when loading a batch of data?*
> >
> > Yes, for non-adaptive subset selection methods, we can parallelize. During the rebuttal time, we parallelized the selection and training phases. Given a budget, we first select a batch of size  b’ < b and then train our model on the batch of size b’. During this training phase, we select and load a different batch of size b’, and keep repeating this procedure so that no additional overhead is spent on subset selection. Here, we set b’=b/10, and thus, used a total of 10 batches all together.
> >
> >
> > Since the subset selection time of both variants of SubSelNet is very small as compared to the training time on the selected subset, we notice no observable improvement in the timing. However, since Facility Location adopts an expensive greedy algorithm to select the subset, parallelization helps in reducing this amount of time significantly. Results of parallel vs pre-training selection using Facility Location on CIFAR-10 for a subset size of 5%.
> >
> >
> > Method|Time (seconds)|Memory (GB-min)
> > -|-|-
> > Parallel|564 |48.275
> > Serial|2981|630.1447667
> >
> > There is no scope for parallelization in the case of adaptive subset selection methods. In the case of GLISTER and Grad-Match, the selection procedure requires access to the model parameters for the selection of the optimal subset. On the other hand, in the case of GraNd and EL2N, we need gradient-based scores to select the subset to train the model further.
> >
> >
> >
> > >*What is the training cost of training the model approximator? Have the authors evaluated the approximator error?*
> >
> >
> > Yes, we already presented the results on model approximation in Figure 6 and Table 8 in the Appendix, which presented (1) a variation of training loss for model approximators with different epochs and (2) a comparison of our model approximator against several alternatives.
> >
> > Specifically, we have evaluated the error between the model approximator and the fully trained models using the KL divergence between output probabilities for different classes given by the model approximator and the trained model. In Appendix C.3, Figure 6, we have reported the variation of the KL divergence $KL(m _{\theta^*}(\mathbf{x} _i) || g _\beta(\mathbf{H} _m, \mathbf{x} _i))$ over 80 epochs. It shows that the KL divergence quickly reduces to ~0.20–0.25 in < 5 epochs for FMNIST and < 20 epochs for CIFAR10.
> > We could fit a model approximator on a 24 GB Titan RTX GPU.
> >
> > In addition, we also performed an ablation study for model approximators in Appendix D.1. Here, we considered different design choices  (Feed-Forward, DeepSet, LSTM, and Transformer) for the architecture of the model approximator, In Table 8, Appendix D.1, we report the KL divergence values between output probabilities for different classes given by the model approximator and the trained model. We quote the table again for convenience.
> >
> > | Model Approximator | CIFAR-10 | FMNIST |
> > |-|-|-|
> > | Feed-Forward | 0.171 | 0.124 |
> > | DeepSet Function | 0.105 | 0.122 |
> > | LSTM | 0.102 | 0.113 |
> > | Our (Transformer) | 0.089 | 0.109 |
> >
> > We observe that our proposal outperforms the other design choices.
> >
> >
> > During the rebuttal period, we performed an extensive ablation study that considers three different design choices of model approximator (Our (Transformer), feedforward, and LSTM) along with three different subset selection strategies (Our subset sampler, top-b instances based on uncertainty and top-b based on loss) which result in nine different combinations of model approximation and subset selection strategies. We measure uncertainty using the entropy of the predicted distribution of the target classes. We report the average test accuracy of the models when they are trained on the underlying pre-selected subset in the following (CIFAR10, 5% subset size)
> >
> >
> > | Design | Accuracy |
> > |-|-|
> > | Feedforward + Our subset sampler | 0.527 |
> > | Feedforward + Uncertainty | 0.379 |
> > | Feedforward + Loss | 0.296 |
> > | LSTM + Our subset sampler | 0.526 |
> > | LSTM + Uncertainty | 0.417 |
> > | LSTM + Loss | 0.283 |
> > | Our (Transformer) + Our subset sampler | 0.548 |
> > | Our (Transformer) + Uncertainty | 0.198 |
> > | Our (Transformer) + Loss | 0.21 |
> >
> >
> > We make the following observations: (1) The complete design of our method, i.e., Our (Transformer) + Our subset sampler performs best. (2) If we use simple unsupervised subset selection heuristics, e.g., loss or uncertainty-based subset selection, our model approximator performs much worse than Feedforward or LSTM. In contrast, this trend is the opposite if we use our subset sampler for selecting the subset. This may be due to overfitting of the transformer architecture in the presence of uncertainty or loss-based selection, which is compensated by our subset sampler.
> >
> > We have added this result in Appendix D.1.

---

> > > ### Comment · Reviewer_a8HM · 2022-12-02
> > > **Reply to Authors of Paper5898**
> > >
> > > Thanks for the detailed response. The author's response addresses some of my concerns, especially the experiments with a wider range of subset ratios.  Thus, I will raise my score.

---

### Official Review · Reviewer_VJSk · 2022-10-25

**Confidence:** 4
**Correctness:** 3
**Technical Novelty And Significance:** 3
**Empirical Novelty And Significance:** 3
**Recommendation:** 6

**Clarity, Quality, Novelty And Reproducibility:**

This paper proposed a new non-adaptive method for the data subset selection problem and has a clear description of the proposed method. Also the experimental results verify the paper’s arguments: mainly on the advantage of the tradeoff between speedup and memory.

**Strength And Weaknesses:**

This paper has a clear decomposition on the model into parts including model approximator and subset sampler. And for each part, it has clear annotations to explain the whole process.
For the subset sampler, 2 variants are proposed and the experimental results show the tradeoff between them, and furthermore, a combination of these 2 variants are tested too.
The comprehensive experimental results proved the effectiveness of the proposed method.
Some questions:
1. some writing errors such as “viz.” appeared a couple of times.
2. both formula (5) and (6) has the E_S, is that correct?
3. for the E_S optimization objective such as (6), as the parameter \pi is under the prob distribution and needs sampling, how do you optimize the \pi? do you use some reparameterization trick which is now shown in the paper.
4. Can you elaborate more on why you don’t jointly optimize the gnn parameter and transformer parameter?
5. In the experimental setup, the paper mentions all the baselines are non-adaptive, is it correct?
6. Do you have any results to show the accuracy gap between the neural model approximator and the fully trained model?


**Summary Of The Paper:**

In  this paper,  a new non-adaptive data subset selection method is proposed. The traditional adaptive method mixed the training and  subset selection. While for the new proposed method, the subset processing is done before the training. Furthermore, the paper also proposed the transductive and inductive variants. The experimental results verified that both variants output perform the baselines on subset selection and also an be used to choose the best architecture.


**Summary Of The Review:**

In summary I think this paper makes a good presentation on a new method for the non-adaptive methods and verifies the contributions in the experimental studies. It will be better to fix some annotations and have more ablation studies and show more results.

---

> ### Author Response · Authors · 2022-11-18
> **Response to Reviewer VJSk (Part-1/2)**
>
> We would like to thank the reviewer for the comments which would improve our paper.
>
>
> > *some writing errors such as “viz.” appeared a couple of times.*
>
> Thank you for pointing them out - we have corrected such grammatical errors in the paper.
>
> > *both formula (5) and (6) has the E _S, is that correct?*
>
> Thank you for catching this mistake. We have rectified it in the paper – equation (5) does not have the expectation over $S$. We quote Eq. (5) again:
>
> $\Lambda(S;m;\pi;F _\phi) = \sum _{i\in S} \ell(F _\phi(G _m, \mathbf{x} _i), y _i) - \lambda H(\Pr(\bullet))$
>
>
> > *for the E _S optimization objective such as (6), as the parameter \pi is under the prob distribution and needs sampling, how do you optimize the \pi? do you use some reparameterization trick which is now shown in the paper.*
>
> At each iteration, we draw instances using $\Pr _{\pi}$ and use a log-derivative trick to compute the gradient of the objective. During each computation step, we use one instance of the ranked list to compute the unbiased estimate of the objective in Eq. (6). We mentioned this in Appendix C.3.
>
>
> > *Can you elaborate more on why you don’t jointly optimize the gnn parameter and transformer parameter?*
>
> In principle, GNN should compute the node and edge embeddings for the network *architecture*. Hence, this embedding should depend *only on the structure of the architecture*, i.e., the operations and their interactions in the model, and *not* on the trained model parameters or the dataset. Such embeddings can then also be used for other datasets without additional training. This significantly reduces the recurring computational cost of re-training neural architectures each time for a new dataset of instances and labels. This led us to separately train GNN on the set of model architectures, leaving aside the ground truth-trained model as well as the dataset.
>
> Instead, if we perform a join optimization of the GNN parameters along with the transformer parameters, the representation learned by the GNN  becomes dependent on the dataset D and all the parameters of the trained models $\theta^*$.  This significantly reduces the influence of the structure of the architecture on the learned embedding.  Moreover, we cannot use the same embeddings to represent the same architecture in case of two different datasets. In addition to that, joint training incurs a recurring computational cost of iterating the training set of neural architectures each time for a new dataset.
>
>
> > *In the experimental setup, the paper mentions all the baselines are non-adaptive, is it correct?*
>
> No, this is not correct. We would like to thank the reviewer for pointing out this typo - not all methods are adaptive. We have rectified the statement in the experimental section. For the convenience of the reviewers, we mention the baselines:
>
> (1) Non-adaptive: Facility Location, Pruning
>
> (2) Adaptive: GLISTER, Grad-Match, EL2N, GraNd
>
>
>
> > *Do you have any results to show the accuracy gap between the neural model approximator and the fully trained model?*
>
> Yes, we already presented such results in Figure 6 and Table 8.
>
> Specifically, we have evaluated the error between the model approximator and the fully trained models using the KL divergence between output probabilities for different classes given by the model approximator and the trained model. In Appendix C.3, Figure 6, we have reported the variation of the KL divergence $KL(m _{\theta^*}(\mathbf{x} _i) || g _\beta(\mathbf{H} _m, \mathbf{x} _i))$ over 80 epochs. It shows that the KL divergence quickly reduces to ~0.20–0.25 in < 5 epochs for FMNIST and < 20 epochs for CIFAR10.
>
> In addition, we also performed an ablation study for model approximators in Appendix D.1. Here, we considered different design choices  (Feed-Forward, DeepSet, LSTM, and Transformer) for the architecture of the model approximator, In Table 8, Appendix D.1, we report the KL divergence values between output probabilities for different classes given by the model approximator and the trained model. We quote the table again for convenience.
>
> | Model Approximator | CIFAR-10 | FMNIST |
> |-|-|-|
> | Feed-Forward | 0.171 | 0.124 |
> | DeepSet Function | 0.105 | 0.122 |
> | LSTM | 0.102 | 0.113 |
> | Our (Transformer) | 0.089 | 0.109 |
>
> We observe that our proposal outperforms the other design choices.

---

> > ### Author Response · Authors · 2022-11-18
> > **Response to Reviewer VJSk (Part-2/2)**
> >
> > > *It would be better to [...] have more ablation studies and show more results*
> >
> > During the rebuttal period, we performed an extensive ablation study that considered three different design choices of model approximator (Our (Transformer), feedforward, and LSTM) along with three different subset selection strategies (Our subset sampler, top-b instances based on uncertainty and top-b based on loss) which result in nine different combinations of model approximation and subset selection strategies. We measure uncertainty using the entropy of the predicted distribution of the target classes. We report the average test accuracy of the models when they are trained on the underlying pre-selected subset in the following (CIFAR10, 5% subset size).
> >
> >
> > | Design | Accuracy |
> > |-|-|
> > | Feedforward + Our subset sampler | 0.527 |
> > | Feedforward + Uncertainty | 0.379 |
> > | Feedforward + Loss | 0.296 |
> > | LSTM + Our subset sampler | 0.526 |
> > | LSTM + Uncertainty | 0.417 |
> > | LSTM + Loss | 0.283 |
> > | Our (Transformer) + Our subset sampler | 0.548 |
> > | Our (Transformer) + Uncertainty | 0.198 |
> > | Our (Transformer) + Loss | 0.21 |
> >
> >
> > We make the following observations: (1) The complete design of our method, i.e., Our (Transformer) + Our subset sampler performs best. (2) If we use simple unsupervised subset selection heuristics, e.g.,  loss or uncertainty-based subset selection, our model approximator performs much worse than Feedforward or LSTM. In contrast, this trend is the opposite if we use our subset sampler for selecting the subset. This may be due to overfitting of the transformer architecture in the presence of uncertainty or loss-based selection, which is compensated by our subset sampler.
> >
> >
> >
> > We added this result in Appendix D.1.
> >
> > Moreover, we added several new experiments. Please refer to the response to reviewer a8HM (Part-1) and the response to reviewer B3Rj (Part-1).

---

### Official Review · Reviewer_B3Rj · 2022-10-28

**Confidence:** 4
**Correctness:** 2
**Technical Novelty And Significance:** 3
**Empirical Novelty And Significance:** 3
**Recommendation:** 5

**Clarity, Quality, Novelty And Reproducibility:**

The approach overall is interesting since, they are able to preselect the dataset for the training process. It seems novel in that aspect.
The paper's clarity can be improved - important and interesting parts have been moved to the appendix, whereas the math behind the model could have been explained better.
The paper as presented, is less easy to reproduce - details about the GNN, the embeddings etc are probably missing.

**Strength And Weaknesses:**

The authors have provided the motivation for the problem, and presented the solution to address the problem. They have also evaluated their approach against 6 other approaches, and 3 different datasets, showing the speed up and memory utilization.
The approach overall is interesting since, they are able to preselect the dataset for the training process.

However, there are a few areas that are not clear from the paper.
1. It would have been great if the authors spent more time discussing the pros- and cons- of their graph network. In general, graph networks themselves can be large, slow and memory consuming. It seems like the comparison is performed on the output of the graph network rather than the end-to-end approach.
The details about the GNN and the graph embedding are important.

2. Much of the details, including the step-by-step algorithm and the details are added to the appendix. For instance, the diagram in appendix B and Pseudocode in C, both would have helped understand the paper better, if it was in the main text.

3. Since the approach relies on pre-selecting, it is not clear how the approach is able to avoid overfitting or underfitting. The authors have split the data into train, validation and test sets. Including a report on the accuracy on these datasets, and the time/ computation resources required for these would have been helpful.

4. The loss function (eq 1) and the objective functions (5 and 6) require more explanation. For intance, the authors state in page 3 after eq1 that: "One can use submodular functions (Fujishige, 2005; Iyer, 2015) like Facility Location, graph cut, or Log-Determinants to model DIVERSITY(S)". However, they havent mentioned the approach they have used in the paper.
Later, they mention the use of entropy on the subset sampler H(Prπ(•)) to model the diversity in page 4 after eq 5 and  KL after eq 6. The choice of the functions needs to be elaborated to appreciate the approach better.

**Summary Of The Paper:**

In this paper, the authors present SUBSELNET - a non- adaptive subset selection framework for solving a particular aspect of subset selection problem - improving generalizability of the subset selection approach; with existing methods, the algorithm has to be executed from the beginning for each new model.
The authors introduce an attention based neural approach that uses the graph structure of the architectures, which is then used to build subset samplers. Their approach has 2 variants: transductive and inductive.
They claim that their approach is more efficient than the existing approaches since the subset is chosen at the beginning of the training process, and the entire dataset is not required through the training process.

**Summary Of The Review:**

The approach overall is interesting since, they are able to preselect the dataset for the training process. The authors introduce an attention based neural approach that uses the graph structure of the architectures, which is then used to build subset samplers. Their approach has 2 variants: transductive and inductive. They claim that their approach is more efficient than the existing approaches since the subset is chosen at the beginning of the training process, and the entire dataset is not required through the training process.

Overall, there are a few areas that are not clear from the paper, including the implementation and the evaluation of the approach.

---

> ### Author Response · Authors · 2022-11-18
> **Response to Reviewer B3Rj (Part-1/2)**
>
> We would like to thank the reviewer for the comments which would improve our paper.
>
> > *It would have been great if the authors spent more time discussing the pros- and cons- of their graph network.*
>
> The advantages of the graph neural networks are as follows:
>
> (1)  Message passing between the nodes (which may be the input, output or any of the operations) allows us to generate embeddings which capture the contextual structural information of the node, i.e. the embedding of each node not only captures the operation for that node, but also the operations preceding that node to a large extent.
>
> (2) It has been shown by [a] and [b] that GNNs are as powerful as the Weisfeiler-Lehman algorithm and thus give a powerful representation for the graph. Thus, we obtain smooth embeddings of the nodes/edges that can effectively distil information from its neighborhood without significant compression.
>
> (3) GNNs embed model architecture into representations independent of the underlying dataset and the model parameters. This is because it operates on only the nodes and edges— the structure of the architecture and does not use the parameter values or input data.
>
> [a] Morris, C., Ritzert, M., Fey, M., Hamilton, W., Lenssen, J., Rattan, G., & Grohe, M.. (2018). Weisfeiler and Leman Go Neural: Higher-order Graph Neural Networks.
>
> [b] Xu, K., Hu, W., Leskovec, J., & Jegelka, S.. (2018). How Powerful are Graph Neural Networks?.
>
>
> In the following, we describe the key disadvantage of GNN.
>
> GNN uses a symmetric aggregator for message passing over node neighbors to ensure that the representation of any node should be invariant to a permutation of its neighbors. Such a symmetric aggregator renders it a low-pass filter, as shown in [c], which attenuates important high-frequency signals.
>
> [c] NT, H.; and Maehara, T. 2019. Revisiting graph neural networks: All we have is low-pass filters.
>
>  We have added this discussion in Appendix E.
>
> > *In general, graph networks themselves can be large, slow and memory consuming.*
>
> On average, each cell of an architecture in the set of neural architectures has ~7 nodes and ~8 edges. Thus, the individual graph sizes are not large, and we could afford to fit a moderate model. Thus we do not face the bottleneck of large memory consumption. Moreover, recent ML libraries dealing with graphs, such as PyTorch-Geometric and Deep Graph Library, are performance-optimized to deal with large graphs; thus memory or time would not be a major issue. However, GNNs also have other disadvantages, which we described above.
>
> > *Much of the details, including the step-by-step algorithm and the details are added to the appendix.*
>
> We have now added the pseudocode for our approach and the details of subroutines in the main text.
>
> > *Since the approach relies on pre-selecting, it is not clear how the approach is able to avoid overfitting or underfitting.*
>
>
> This is indeed an excellent point. Yes, since we train with a small amount of data, there is a possibility of overfitting. However, the coefficient $\lambda$ of the entropy regularizer $\lambda H( Pr _{\pi} )$ can be increased to draw instances from the different regions of the feature space, which in turn can reduce overfitting. In practice, we tuned $\lambda$ on the validation set to control such overfitting.
>
> We present the accuracies on (training, validation, and test) folds for both transductive and inductive models on CIFAR10.
>
> **Accuracies on training data**
> Subset Size(%) | Transductive | Inductive
> - | - | -
> 10|0.728|0.66
> 20|0.852|0.673
> 40|0.89 |0.691
> 70|0.942|0.738
>
> **Accuracies on validation data**
>
>
> Subset Size(%) | Transductive | Inductive
> - | - | -
> 10|0.702|0.632
> 20|0.809|0.658
> 40|0.856|0.678
> 70|0.912|0.717
>
> **Accuracies on test data**
>
> Subset Size(%) | Transductive | Inductive
> - | - | -
> 10|0.678|0.606
> 20|0.77 |0.644
> 40|0.825|0.666
> 70|0.884|0.698
>
> We make the following observations: (1) From training to test, in most cases, the decrease in accuracy is~7%. (2) This small accuracy gap is further reduced from validation to test. Here, in most cases, the decrease in accuracy is ~4%. We perform early stopping using the validation set, which acts as an additional regularizer; therefore, the amount of overfitting is significantly low.
>
>  We have added this discussion in Appendix D.3.

---

> > ### Author Response · Authors · 2022-11-18
> > **Response to Reviewer B3Rj (Part-2/2)**
> >
> > > *The loss function (eq 1) and the objective functions (5 and 6) require more explanation. For intance, the authors state in page 3 after eq1 that: "One can use submodular functions (Fujishige, 2005; Iyer, 2015) like Facility Location, graph cut, or Log-Determinants to model DIVERSITY(S)". However, they havent mentioned the approach they have used in the paper. Later, they mention the use of entropy on the subset sampler H(Prπ(•)) to model the diversity in page 4 after eq 5 and KL after eq 6. The choice of the functions needs to be elaborated to appreciate the approach better.*
> >
> >
> > In Eq (1), we introduced the original subset selection problem where optimization variable S— the subset of instances— makes the underlying problem combinatorial. Here, our key message is that one can use submodular functions like Graph-Cut, Facility-Location, and Log-Determinant as the diversity functions, which would allow us to use greedy algorithms to maximize the objective in Eq. (1). But, as we discussed, this suffers from two bottlenecks: expensive computation issues and lack of generalizability. Therefore, we do not follow these approaches and resort to our proposed approach called SubSelNet. Please check the para with the heading “Bottlenecks of the combinatorial optimization” after the para following Eq (1)).
> >
> >
> > In contrast to the optimization problem in Eq. (1) which was a combinatorial set optimization problem, the optimization problem in SubSelNet in Eq. (6) is a continuous optimization problem where the goal is to estimate $\Pr_{\pi}$. Here, the probability distribution is the key optimization variable, and therefore, entropy is a more natural measure of diversity than the other submodular measures.  We added this discussion in the revised version in Appendix F.
> >
> > Note that the KL term in Eq. (6) has nothing to do with diversity.  Note that one key component of our model is the model approximator $F_{\phi}$, whose goal is to predict the output of the trained model. To this goal, we minimize the KL divergence between the output probabilities from $F_{\phi}$ and the probabilities given by the (gold) trained model $m_{\theta^*}$. Thus the purpose of including the KL term is targeted training of $F_{\phi}$.
> >
> >
> > > *details about the GNN,is probably missing*
> >
> > For convenience, we quote the GNN model (Eqs (8,9) in our paper) here:
> >
> > $ h _{u}[0] = \text{InitNode} _{\alpha}(f_u)$,
> >
> > $ h _{(u,v)}[k-1] = \text{EdgeEmbed} _{\alpha}(h _u[k-1],h _v[k-1])$
> >
> > $h' _{u}[k-1] = \text{SymmAggr} _{\alpha} (\{h _{(u,v)}[k-1] | v \in \text{Nbr}(u)})$
> >
> > $h’ _u[k] = \text{Update} _{\alpha}(h _u[k-1], h' _{u}[k-1]) ,  k<K$
> >
> >
> > The training of the network has been stated in Section 4.2, and the implementation details about GNN were already added in Appendix C.3. We have further elaborated on it here as well as in the revised version (Appendix C.3)
> >
> > For each architecture, we use the initial feature $f_u \in R^5$ in Eq. (8) as a five-dimensional one-hot encoding for each operation. This is fed into $\text{InitNode}$ (Eq. 8)  to obtain a 16-dimensional output. Here, $\text{InitNode}$  consists of 5 x 16 linear, ReLU and 16 x 16 linear layers cascaded with each other. Each of  $\text{EdgeEmbed}$ and $\text{Update}$ consists of a 5 x 128 linear-BatchNorm-ReLU cascaded with a 128 x 16 linear layer. Moreover, the symmetric aggregator is a sum aggregator.
> >
> > In our implementation, we set the number of recursive layers $K = 5$, the number of epochs to 10, the batch size to 32, and use the Adam optimizer with a learning rate of 1e-3 in our training mechanism.
> >
> >
> > [d] ​​You, J., Ying, R., Ren, X., Hamilton, W., & Leskovec, J.. (2018). GraphRNN: Generating Realistic Graphs with Deep Auto-regressive Models.

---

### Decision · Program_Chairs · 2023-01-20

**Decision:**

Reject

**Justification For Why Not Higher Score:**

experimental results are not convincing because it is validated only in regimes where performance is sacrificed too much

**Justification For Why Not Lower Score:**

n/a

**Metareview: Summary, Strengths And Weaknesses:**

This paper proposes a subset selection method that embeds the neural network architecture via GNNs and uses it as a surrogate for trained deep NNs. Three reviewers reviewed this paper. Two reviewers suggested borderline accept, and one suggested borderline reject. Through rebuttal, the authors somewhat improved the clarity on some details such as GNN architecture, which was one of the main reasons for suggesting rejection. However, concerns about overfitting due to non-adaptive characteristics still remain.

At the request of the authors, this AC also carefully reviewed this paper, but I also suggest rejecting this paper for a different reason than the negative reviewer. (This is an issue raised by other positive reviewer.) It is important to secure computational efficiency through subsampling, but it is meaningless if too much performance is sacrificed. In other words, it is necessary to verify the proposed method on a much more diverse subset size than that shown in figure 1. (Especially, verification in a region where performance is not that sacrificed is important.) During the rebuttal period, the authors reported experiments on more diverse subset sizes, but only marginal improvements over baselines were achieved. In addition, unlike baseline papers that verified various architectures such as ResNet and MobileNet, the experiment was verified only in limited architectures on CIFAR10 in the rebuttal, so I think that this part needs more verifications and an additional round of review.